# BiQAP: Neural Bi-level Optimization-based Framework for Solving Quadratic Assignment Problems

## Abstract

Quadratic Assignment Problem (QAP) has attracted lasting attention for its wide applications and computational challenge. Despite the rich literature in machine learning for QAP, most works often address the problem in the setting of image matching, whereby deep networks could play a vital role in extracting useful features for the subsequent matching. While its power on pure numerical QAP instances is limited in node embedding, often with a vanilla graph neural network. This paper tries to tap the potential of deep nets for QAP, specifically by modifying the input instance which is orthogonal to previous efforts. Specifically, we develop a bi-level unsupervised framework, where the inner optimization involves trying to solve the modified instance with entropic regularization that can be solved iteratively using the Sinkhorn algorithm without affecting backpropagation by truncating gradients during training. The outer minimization deals with the quadratic objective function of the original QAP. In particular, seeing the intractable scale of the most general form i.e. Lawler's QAP and the practical utility of the more efficient Koopmans-Beckmann QAP (KBQAP) form for solving other graph and combinatorial problems like TSP and graph edit distance, we embody our network on the KBQAP, and show its strong performance on various benchmarks in our experiments. Source code will be made publicly available.

## 1 Introduction

The quadratic assignment problem (QAP) (Koopmans & Beckmann, 1957) is one of the fundamental combinatorial optimization problems known in general NP-hard and many classic problems can be formulated in a QAP form such as facilities location problems (Owen & Daskin, 1998), graph matching tasks (Livi & Rizzi, 2013), graph edit distance (Sanfeliu & Fu, 1983) and traveling salesman problems (Gutin & Punnen, 2006). The most general form of QAP is called Lawler's QAP (LLQAP) (Lawler, 1963):

$$\max_{\mathbf{X} \in \{0,1\}^{n_1 \times n_2}} J(\mathbf{X}) = \text{vec}(\mathbf{X})^\mathsf{T} \mathbf{K} \text{vec}(\mathbf{X}), \text{ s.t. } \mathbf{X}\mathbf{1}_{n_2} = \mathbf{1}_{n_1}, \mathbf{X}^\mathsf{T}\mathbf{1}_{n_1} \leq \mathbf{1}_{n_2}, n_1 \leq n_2 \quad (1)$$

where $\mathbf{K} \in \mathbb{R}^{n_1 n_2 \times n_1 n_2}$ represents the affinity matrix. One of the most popular applications of LLQAP is graph matching (Yan et al., 2016), whereby the edge-wise similarity (stored in the off-diagonal elements of $\mathbf{K}$) and node-wise affinity (stored in the diagonal elements of $\mathbf{K}$) are both incorporated in the overall quadratic objective function for maximization. The marginal constraints are often called permutation or matching constraints in graph matching literature. Despite its universality, one practical drawback is that the above LLQAP has to carry the burdensome affinity matrix $\mathbf{K}$ which significantly restricts its applicability for modelling the real-world problems, usually only up to dozens of nodes. This circumstance compels researchers and practitioners to adopt a more lightweight formulation regarding with the space complexity by avoiding explicitly storing $\mathbf{K}$, namely the well-known Koopmans-Beckmann QAP (KBQAP) (Koopmans & Beckmann, 1957):

$$\max_{\mathbf{X} \in \{0,1\}^{n_1 \times n_2}} J(\mathbf{X}) = \text{tr}(\mathbf{X}^\mathsf{T} \mathbf{F}_1 \mathbf{X} \mathbf{F}_2) + \text{tr}(\mathbf{K}_p^\mathsf{T} \mathbf{X}), \text{ s.t. } \mathbf{X}\mathbf{1}_{n_2} = \mathbf{1}_{n_1}, \mathbf{X}^\mathsf{T}\mathbf{1}_{n_1} \leq \mathbf{1}_{n_2}, n_1 \leq n_2 \quad (2)$$

where $\mathbf{F}_1 \in \mathbb{R}^{n_1 \times n_1}$ and $\mathbf{F}_2 \in \mathbb{R}^{n_2 \times n_2}$ are weighted adjacency matrices for edges, and $\mathbf{K}_p \in \mathbb{R}^{n_1 \times n_2}$ is node-to-node affinity matrix. Many classic combinatorial problems, e.g. Graph Matching (GM), TSP, and Graph Edit Distance (GED), can be readily rewritten by KBQAP.

Along the emerging trend of machine learning for combinatorial optimization (Bengio et al., 2021), this paper aims to develop a new learning paradigm to address this fundamental problem. In fact, a widely used technique for handling the permutation constraints in Eq. 2 is the Sinkhorn layer that enforces the given matrix to become a doubly-stochastic one, as such the final feasible solution could be readily obtained via solving a linear assignment problem by e.g. the Hungarian method (Kuhn, 1955). In this way, solving the QAP problem either in LLQAP (Lawler, 1963) or KBQAP (Koopmans & Beckmann, 1957) form could be fulfilled by an inference step with a neural network to fully utilize the parallel GPU computing resource. Moreover, the training could also be performed in an end-to-end fashion either by supervised or unsupervised learning.

In our investigation of the most common graph matching approaches (Jiang et al., 2022; Wang et al., 2023) in QAP, we discover that many of them conduct experiments on visual datasets like Willow (Cho et al., 2013), where annotations are typically manually crafted as supervised data for training Graph Neural Networks (GNNs) (Zhou et al., 2020) to learn the mapping. However, we raise concerns regarding this practice since the ground-truth matching in supervised data may not necessarily be the optimal solution in the context of QAP optimization. To illustrate this point, we present an example in Fig. 1 where visually incorrect matches yield a better objective function in Eq. 2. Consequently, visual graph matching may significantly differ from solving QAP in the field of combinatorial optimization. This discrepancy often arises from a focus on image feature extraction rather than addressing the core principles of combinatorial optimization.

Building upon the above reconsideration, this paper places a heightened focus on learning a better QAP objective function based on the parameters outlined in Eq. 2, rather than being concerned with whether nodes or edges are correctly matched visually. We aim to train a neural network-based mapping that takes the QAP formula's essential matrices ($\mathbf{K}_p$, $\mathbf{F}_1$, $\mathbf{F}_2$) as input and produces the corresponding optimal solution $\mathbf{X}$ that maximizes the objective function in Eq. 2. Consequently, our model exhibits enhanced versatility, transcending the confines of a specific task, as it can be applied to derive solutions as long as the task can extract the key matrices relevant to the QAP.

Different from previous works employing Sinkhorn layers to get the doubly stochastic matrix as output, we introduce a novel framework called BiQAP with bi-level optimization, which maps the original QAP formula to another new optimization one, specifically an entropic regularized QAP in the inner optimization. Note a differentiable approximate solver known as the Gromov-Wasserstein Sinkhorn (GW-Sinkhorn) (Peyré et al., 2016) algorithm is utilized as a layer to solve the entropic regularized QAP optimization. Similar to existing QAP solvers, the GW-Sinkhorn algorithm is susceptible to local optima. Therefore, during the training phase, we initialize the GW-Sinkhorn algorithm with multiple Gumbel samples, compelling the outputs to yield solutions $\mathbf{X}$ that minimize the original QAP objective function in the outer optimization. This approach aims to mitigate the impact of local optima. **The highlights of this work include:**

1) We propose an efficient neural QAP framework called BiQAP under a bi-level optimization paradigm. The outer optimization corresponds to the objective function of the original QAP, while the inner minimization is conducted via an iteratively learned QAP with entropic regularization, in which Gromov-Sinkhorn algorithm is adopted as the differential approximate QAP solver to obtain the solution. In contrast to peer learning-based methods, our end-to-end approach is capable of producing a high-quality solution without heavily relying on extensive random sampling techniques (Wang et al., 2021a) or intricate post-processing algorithms (Piao et al., 2023).

2) Note our BiQAP focuses on optimizing the general form of QAP without requiring the explicit input node/edge features as widely used and learned from existing learning of QAP works, which typically formulate the problem as a graph matching task (Yan et al., 2020). In contrast, BiQAP is concerned solely with the optimization formula without considering learning of the input features which we believe to some extent distract the solving of QAP itself.

3) To neuralize the QAP, we present FormulaNet, where the crucial matrices of the optimization formula serve as inputs and the crucial matrices of another optimization problem are generated as outputs. Our FormulaNet can accommodate QAP problems of various sizes.

4) We conduct extensive experiments across five typical QAP-based tasks. Specifically, our method achieves state-of-the-art performance in the Graph Matching, Large Random QAP optimization, and Graph Edit Distance tasks, outperforming both learning-free and learning-based approaches. Meanwhile, it delivers competitive results in the Traveling Salesman Problem and QAPLIB tasks.

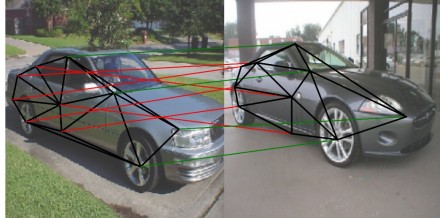

(a) Ground truth matching with objective 35.77.   (b) Optimal matching by Gorubi with objective 38.44.

Figure 1: Graph matching test on an instance. The QAP instance is extracted by a pre-trained module from Wang et al. (2021a). We calculate the objective for both the ground truth and the solution obtained by Gurobi. The left shows the correct matching, while the right image displays the optimal solution found by Gurobi. The visually incorrect matches on the right achieve a better objective value.

## 2 RELATED WORKS

**QAP Solvers and Corresponding Tasks.** Due to its significance, there is a wealth of work on the QAP. Learning-free methods for LLQAP (Hahn et al., 1998; Leordeanu & Hebert, 2005; Leordeanu et al., 2009; Cho et al., 2010; Wang et al., 2017) and KBQAP (Edwards, 1980; Erdoğan & Tansel, 2007; Kushinsky et al., 2019) typically search for a feasible solution based on specific settings. In recent years, with the rise of neural networks, several learning-based methods have emerged, mostly from the perspective of graph matching. KBQAP solvers (Nowak et al., 2018; Wang et al., 2019; Yu et al., 2019) rely on structured graph inputs, limiting their ability to handle arbitrary QAP instances. The LLQAP solver NGM (Wang et al., 2021a) extracts graph information into a matrix $\mathbf{K}$ via convolutional layers, but its $O(n^4)$ complexity leads to significant memory and GPU limitations on large-scale problems. In contrast, our learning-based BiQAP supports generalized inputs for any QAP formulation and shows superior capability in handling large-scale instances due to its KBQAP foundation. QAP-related tasks are diverse in previous research, including graph matching (Yan et al., 2016), traveling salesman problem (Gutin & Punnen, 2006; Ye et al., 2023), and QAPLIB (Burkard et al., 1997). Beyond these, we innovatively formulate the graph edit distance (Sanfeliu & Fu, 1983) as a QAP problem, achieving state-of-the-art results, and construct an extremely large, randomly generated dataset to evaluate the model's ability to solve large-scale problems.

**Gromov-Wasserstein Distance.** The Wasserstein Distance (Le et al., 2019) is employed to compare probability distributions, typically represented as histograms in finite-dimensional spaces for optimal transport (Peyre & Cuturi, 2019; Shi et al., 2024b), either within the same ground space or across pre-registered ground spaces. In contrast, the Gromov-Wasserstein Distance (Mémoli, 2014) extends the concept to cases where ground spaces are not pre-registered, necessitating a non-convex quadratic program (Xia et al., 2015) to compute the transport, resulting in a soft registration between domains. Our algorithm is inspired by the computation of the Entropic Regularized Gromov-Wasserstein Distance (Peyré et al., 2016), utilizing a mirror-descent scheme based on matrix iterations to solve this problem. We account for the asymmetry present in Gromov-Wasserstein Distance matrices and leverage this differentiable algorithm as a pivotal layer within the end-to-end BiQAP framework.

**Designing the loss via Bi-level Optimization.** Bi-Level Optimization is originated from economic game theory (Fortuny-Amat & McCarl, 1981) and then introduced into the optimization community (Dempe, 2020), which handle problems with a hierarchical structure, involving two levels of optimization tasks, where one task is nested inside the other. Despite the different motivations and mechanisms in machine learning, a lot of complex problems, such as neural architecture search (Liu et al., 2018), adversarial learning (Li et al., 2019) and deep reinforcement learning (Zhang et al., 2020), actually all contain a series of closely related subproblms. In this paper, we mainly follow (Shi et al., 2023) that understanding or designing the loss via bi-level optimization:

$$\min_{\theta} KL(\tilde{\mathbf{P}} \mid \mathbf{P}^{\theta}) \quad \text{s.t.} \quad \mathbf{P}^{\theta} = \arg\min_{\mathbf{P1}=\mathbf{1}} \langle \mathbf{C}^{\theta}, \mathbf{P} \rangle - \epsilon H(\mathbf{P}), \tag{3}$$

where $\mathbf{C}^{\theta}$ represents the cosine distance for features with parameters $\theta$, and $\tilde{\mathbf{P}}$ is the known supervision for learning. As proven in (Shi et al., 2023), $H(\mathbf{P}) = -\langle \mathbf{P}, \log \mathbf{P} - \mathbf{1} \rangle$ is the entropic regularization with coefficient $\epsilon$. The inner optimization is exactly equivalent to the softmax activation, while the outer optimization corresponds to cross-entropy. Thus, the entire bi-level optimization is equivalent to the InfoNCE loss, with $\epsilon$ acting as the temperature in softmax. (Shi et al., 2023; 2024a)

proposed modifying the inner optimization to define a new loss. In this paper, we follow these studies and use a bi-level optimization approach to design the loss for our BiQAP model, where the outer optimization adopts the original QAP objective and the inner optimization uses the learned entropic-regularized QAP.

## 3 METHODOLOGY

### 3.1 OVERVIEW: UNSUPERVISED LEARNING OF QAP WITH BI-LEVEL OPTIMIZATION

Solving the QAP problem has always been a challenge, especially for large instances. It is difficult to obtain accurate solutions within a limited time due to the tendency of most algorithms to get stuck in local optima. Traditional heuristic algorithms (Held & Karp, 1970; Riesen et al., 2007) are mostly based on search methods, resulting in high time complexity, especially when dealing with large-scale instances. Apart from search-based algorithms, the current trend leans towards GPU-friendly learning-based algorithms based on neural networks (Wang et al., 2021a) or matrix iterative algorithms (Kushinsky et al., 2019). These algorithms typically learn probability matching matrices and obtain solutions through Hungarian post-processing (Kuhn, 1955). While these algorithms may not match the precision of the former (i.e., search-based algorithms), they excel in batch computations, offering a time advantage. In this paper, following the trend of the latter type of work (Wang et al., 2021a), a new GPU-friendly QAP solver is proposed for solving KBQAP, which combines the strengths of neural networks and matrix iterative algorithms. It has already shown strong competitiveness over both heuristic and learning-based algorithms.

Diverging from previous works, to solve the QAP problem as given in Eq. 2, we introduce a new concept that transforms the old problem into a new one less affected by local optima, allowing us to obtain the solution to the original problem by solving the equivalent new problem. Specifically, we propose BiQAP into the form of bi-level optimization:

$$\min_{\theta} -\mathrm{tr}\big((\mathbf{X}^{\theta})^{\mathsf{T}}\mathbf{F}_1\mathbf{X}^{\theta}\mathbf{F}_2\big) - \mathrm{tr}(\mathbf{K}_p^{\mathsf{T}}\mathbf{X}^{\theta})$$

$$\text{s.t.} \quad \mathbf{X}^{\theta} = \arg\min_{\mathbf{X}\mathbf{1}_{n_2}=\mathbf{1}_{n_1}, \mathbf{X}^{\mathsf{T}}\mathbf{1}_{n_1}\leq\mathbf{1}_{n_2}} -\mathrm{tr}(\mathbf{X}^{\mathsf{T}}\mathbf{F}_1^{\theta}\mathbf{X}\mathbf{F}_2^{\theta}) - \mathrm{tr}\big((\mathbf{K}_p^{\theta})^{\mathsf{T}}\mathbf{X}\big) - \epsilon H(\mathbf{X})$$

(4)

where $H(\mathbf{X}) = -\langle\mathbf{X}, \log\mathbf{X} - \mathbf{1}_{n_1\times n_2}\rangle$ is the entropic regularization with regularization coefficient $\epsilon$ and $(\mathbf{F}_1^{\theta}, \mathbf{F}_2^{\theta}, \mathbf{K}_p^{\theta}) = f_{\theta}(\mathbf{F}_1, \mathbf{F}_2, \mathbf{K}_p)$ denotes the new QAP instance learned by neural network $f_{\theta}$. In the above bi-level optimization, we transform the original QAP into a new entropy-regularized QAP using the neural network $f_{\theta}$, and obtain the solution $\mathbf{X}^{\theta}$ through a matrix iterative algorithm. The neural network parameters $\theta$ are then optimized to minimize the original problem. One can understand our QAP learning framework in Eq. 4 based on Eq. 3. Specifically, in the inner optimization, we first input the original QAP parameters $\mathbf{F}_1, \mathbf{F}_2$, and $\mathbf{K}_p$ into the neural network $f_{\theta}$ to obtain a new entropic regularized QAP with parameters $\mathbf{F}_1^{\theta}, \mathbf{F}_2^{\theta}$, and $\mathbf{K}_p^{\theta}$. We then apply the matrix iterative algorithm proposed in (Peyré et al., 2016) as a differentiable solver to solve the new entropic regularized QAP and obtain the solution $\mathbf{X}^{\theta}$. Note that the differentiable solver acts as an activation layer similar to softmax or sinkhorn, allowing gradient backpropagation. Finally, given the calculated solution $\mathbf{X}^{\theta}$, we minimize the negative objective of the original QAP, which serves as the loss function. The details are discussed in the next subsection.

### 3.2 UNSUPERVISED LEARNING FOR QUADRATIC ASSIGNMENT PROBLEMS

**FormulaNet: Embedding the QAP Formula to Neural Networks.** The overall framework is illustrated in Fig. 2 and the algorithm is shown in Alg. 1. For the FormulaNet, various structures can be employed to transform the original KBQAP instance into a new one, as long as they meet the following criteria: 1) They can accept matrices of arbitrary shapes as input; 2) The output matrix retains the same shape as the input. In our model, we utilize the Mamba architecture (Gu & Dao, 2023) as the **FormulaNet**, a structure that has gained recent attention (Zhu et al., 2024). Its key advantage of linear scaling with sequence length allows for improvements in efficiency while meeting our requirements. In contrast, Attention-based models such as the Vision Transformer (ViT) (Dosovitskiy, 2020) have quadratic complexity, which leads to excessive computational and memory costs when handling large-scale problems. Moreover, we do not adopt Graph Neural Networks (GNNs), which are widely used in combinatorial optimization research (Wang et al., 2020a; 2021c), as GNNs are better suited for structured inputs like graphs, whereas our input matrices lack clear structural

Figure 2: Structure of our proposed BiQAP. On the left, it learns an new QAP instance through the learnable FormulaNet. On the right, the differentiable QAP solver leverages the Gromov-Sinkhorn algorithm to approximately solve the new instance. We use an unsupervised loss to perform back-propagation during the training stage.

patterns. Other architectures, such as Linear Attention (Wang et al., 2020b), can also serve as the FormulaNet for BiQAP. In this paper, the use of Mamba is an initial choice, leaving room for future exploration and refinement in the design of the FormulaNet. Additional details of the structure are discussed in Appendix E. In BiQAP, each of the original KBQAP instances, $\mathbf{F}_1$, $\mathbf{F}_2$, and $\mathbf{K}_p$, is processed through this FormulaNet, resulting in new instance with its key matrices $\mathbf{F}_1^\theta$, $\mathbf{F}_2^\theta$, and $\mathbf{K}_p^\theta$.

**Differentiable QAP Approximate Solver: Gromov-Sinkhorn.** Now, given a new QAP instance with parameters $\mathbf{F}_1^\theta$, $\mathbf{F}_2^\theta$, and $\mathbf{K}_p^\theta$, we need to solve the problem. However, using a heuristic algorithm like A* search (Riesen et al., 2007) inhibits gradient backpropagation, making end-to-end learning impossible and requiring methods like reinforcement learning to update parameters $\theta$, which is not conducive to learning a more generalized neural QAP solver. Even if backpropagation were feasible, traditional heuristic algorithms tend to be time-inefficient, resulting in slow model training. In this paper, we propose an efficient QAP solver as a large activation layer to enhance model training. We relax the original 0-1 constraints and modify the optimization equation using entropy regularization:

$$\min_{\mathbf{X}} -\mathrm{tr}(\mathbf{X}^\mathsf{T}\mathbf{F}_1^\theta\mathbf{X}\mathbf{F}_2^\theta) - \mathrm{tr}(\mathbf{K}_p^{\theta\mathsf{T}}\mathbf{X}) - \epsilon H(\mathbf{X}), \text{ s.t. } \mathbf{X}\mathbf{1}_{n_2} = \mathbf{1}_{n_1}, \mathbf{X}^\mathsf{T}\mathbf{1}_{n_1} \leq \mathbf{1}_{n_2} \quad (5)$$

which is exactly the inner optimization in Eq. 4. To solve the above optimization, one can use iteratively Sinkhorn algorithm to progressively compute a stationary point, as specified by:

$$\mathbf{X}^{(l)} = \arg\min_{\mathbf{X}} \left\langle \mathbf{C}^{(l)}, \mathbf{X} \right\rangle - \epsilon H(\mathbf{X}), \text{ s.t. } \mathbf{X}\mathbf{1}_{n_2} = \mathbf{1}_{n_1}, \mathbf{X}^\mathsf{T}\mathbf{1}_{n_1} \leq \mathbf{1}_{n_2},$$

$$\text{where} \quad \mathbf{C}^{(l)} = -\mathbf{F}_1^\theta\mathbf{X}^{(l-1)}\mathbf{F}_2^\theta - \mathbf{F}_1^{\theta\mathsf{T}}\mathbf{X}^{(l-1)}\mathbf{F}_2^{\theta\mathsf{T}} - \mathbf{K}_p^\theta. \quad (6)$$

---

**Algorithm 1:** Training and Inference of BiQAP

1 **Input:** Original problem instance inputs $\mathbf{F}_1, \mathbf{F}_2, \mathbf{K}_p$
2 $\mathbf{F}_1^\theta, \mathbf{F}_2^\theta, \mathbf{K}_p^\theta$ = FormulaNet($\mathbf{F}_1, \mathbf{F}_2, \mathbf{K}_p$);
3 $\mathbf{X}^{(0)} \sim \mathrm{Gumbel}^{n1 \times n2}$; // initialization
4 **for** $l = 1, 2, \ldots, L$ **do**
5 $\quad$ construct $\mathbf{C}^{(l)}$ by Eq. 6; // stationary point
6 $\quad$ $\mathbf{P} = \exp\left(-\mathrm{norm}(\mathbf{C}^{(l)})/\epsilon\right)$;
7 $\quad$ **repeat**
8 $\quad\quad$ $\mathbf{P} = \mathrm{diag}\left((\mathbf{P}\mathbf{1}_n)^{-1}\right)\mathbf{P}$;
9 $\quad\quad$ $\mathbf{P} = \mathbf{P}\mathrm{diag}\left(\min\left((\mathbf{P}^\mathsf{T}\mathbf{1}_m)^{-1}, \mathbf{1}_n\right)\right)$;
10 $\quad$ **until** *convergence*;
11 $\quad$ $\mathbf{X}^{(l)} = \mathbf{P}$;
12 **if** *training* **then**
13 $\quad$ **return** $\mathbf{X}^{(L)}$; // differentiable for calculating loss
14 **else**
15 $\quad$ **return** Hungarian($\mathbf{X}^{(L)}$);

---

By initializing $\mathbf{X}^{(0)}$, we can compute $\mathbf{C}^{(1)}$, and then optimizing the entropic regularized OT (Sinkhorn, 1964) (i.e., running the Sinkhorn algorithm) to obtain $\mathbf{X}^{(1)}$. Similarly, we iteratively compute $\mathbf{C}^{(l)}$ and $\mathbf{X}^{(l)}$ alternately until convergence. A detailed proof of Eq. 6 is provided in Appendix D. Next, we will discuss how to make a new QAP instance approximate the original one and explore how our method helps the obtained results escape local optima, which is a challenge faced by all non-convex or combinatorial optimizations.

**Gumbel Sampling-based Unsupervised Loss.** Like original QAP instance, the obtained new QAP instance inevitably involves the issue of local optima dependence on initialization.

How to reduce local optima and make the obtained solution approximate the original QAP instance is the central challenge in our study. Given that the Gromov-Sinkhorn algorithm relies on initialization, we perform multiple samplings of initializations, aiming for each sampled result to approximate the optimal solution of the original problem. Consequently, during the inference process, when presented with a testing QAP instance, we no longer need to seek a better initialization or employ momentum to escape local optima. Due to the inherent difficulty of finding the optimal solution for QAP itself, especially when dealing with large scales, we focus on optimizing the objective of the original QAP instance:

$$\mathcal{L} = \mathbb{E}_{z \sim \mathbb{P}_G} - \text{tr}(\mathbf{X}_z^\mathsf{T} \mathbf{F}_1 \mathbf{X}_z \mathbf{F}_2) - \text{tr}(\mathbf{K}_p^\mathsf{T} \mathbf{X}_z), \tag{7}$$

where $\mathbb{P}_G$ is the Gumbel distribution and $\mathbf{X}_z$ is the solution iterated by Eq. 6 given the Gumbel sample $z$ as initialization of $\mathbf{X}^{(0)}$.

**Inference Method.** Then, given a new QAP formula, we can perform inference using the trained neural networks (i.e., FormulaNet) and the QAP approximate Solver to obtain the doubly stochastic matrix. Across various experimental tasks, the quality of its solutions is very high, and we can obtain the final exact integer solution through a simple application of the Hungarian algorithm.

### 3.3 FURTHER DISCUSSIONS

**Comparisons to NGM (Wang et al., 2021a) in our view.** To the best of our knowledge, NGM (Wang et al., 2021a) is the first work to utilize neural networks to solve QAP. However, based on our experiments, NGM relies heavily on numerous Gumbel repeated samplings, and if the number of samplings is reduced, its performance deteriorates significantly, which differs from our BiQAP as shown in the experiments in Fig. 3. From the perspective of our optimization problem transformation, considering the implicit optimization problem in the Sinkhorn algorithm, NGM actually transforms QAP into an entropic optimal transport problem, which is exactly a convex problem and may be not complex enough to fit a more complicated QAP. Thus, we cannot solely consider the output of the constraints, but also consider their implicit optimization problem.

**Difference to Other Graph-based Models.** Here we want to emphasize the difference between our work and other graph-based works in that we are no longer studying a specific task, e.g., graph matching (Wang et al., 2020a), but rather aim to learn the mapping between different optimization problems and their solutions. Thus, given different optimization formula, we can quickly obtain corresponding solutions through neural networks instead of complex optimization algorithms, e.g., simplex method (Dantzig, 1951), interior point method (Karmarkar, 1984), etc. The success of unsupervised learning in this work gives us confidence that we can extend our research to other convex or non-convex problems, and we believe this will have a significant impact on combinatorial optimization and operations research.

**Further Discussion on Differentiable Approximate Solver.** In fact, the differentiable approximate solver as a layer is a crucial component in our model that can optimize the objective function of the new entropic QAP while satisfying constraints (doubly stochastic matrix). When we consider solving other optimization problems, e.g., linear, quadratic, or other non-convex optimization problems, how to select a new transformed optimization problem to fit as many original optimization problems as possible and solve them through matrix iteration is an area that needs to be explored in the future. Besides, combining with traditional algorithms such as interior point methods may further improve the prediction results.

## 4 EXPERIMENT

Experiments are conducted on a Linux workstation using an NVIDIA GeForce RTX 3090 GPU and an Intel(R) Core(TM) i9-10920X CPU @ 3.50GHz, with programs implemented in *PyTorch*. We evaluate Quadratic Assignment Problems (QAP) in five different tasks as case studies, along with an ablation study, sampling experiments, and generalization tests.

### 4.1 CASE STUDY I: GRAPH MATCHING DATASET

**Protocol setting.** In this experiment, we follow (Wang et al., 2019; Ye et al., 2023) to generate random point sets in a 2D plane to compare with other competitive methods. We first create 10K ground truth points with coordinates sampled from $U(0, 1) \times U(0, 1)$, where $U$ is a uniform distribution,

and perturb them using random scaling from $U(1 - \delta_s, 1 + \delta_s)$ and additive noise from Gaussian $N(0, \sigma_n^2)$. These ground truth points form a target graph, while the distorted points, after randomly permuting node order, form a reference graph. The original graph matching serves as the ground truth solution, though it is often sub-optimal in the KBQAP formulation due to the randomness introduced by the perturbations. We sample two configurations, GM-I and GM-II, with 2K graphs for training and 0.2K for testing, using $(\delta_s, \sigma_n)$ values of $(0.05, 0.02)$ and $(0.3, 0.2)$, respectively. Each graph contains 128 nodes. For the KBQAP formulation, similarity matrices are computed as $\mathbf{T}_{i,j} = \exp(-L_2(c_1, c_2))$, where $c_1$ and $c_2$ are node coordinates from the target or reference graph. The matrix $\mathbf{T}$ can represent $\mathbf{F}_1$, $\mathbf{F}_2$, or $\mathbf{K}_p$, where $\mathbf{F}_1$ and $\mathbf{F}_2$ are the intra-similarity matrices for the target and reference graphs, and $\mathbf{K}_p$ is the inter-similarity matrix between the two graphs.

**Baseline.** Existing methods addressing the KBQAP problem (Nowak et al., 2018; Wang et al., 2019; Yu et al., 2019) do not directly solve the KBQAP problem with three given matrices. Instead, they process the input images or graphs from a graph matching perspective. Here we compare with an efficient learning-free KBQAP method, $\Delta$**-Search**, which employs the concept of 2OPT (Lin & Kernighan, 1973). Additionally, since KBQAP can be transformed into LLQAP without considering computational complexity, we also compare with the LLQAP methods[1]: 1) **SM** (Leordeanu & Hebert, 2005) considers graph matching as finding graph clusters using spectral numerical techniques; 2) **RRWM** (Cho et al., 2010) adopts random-walk to match nodes in a graph pair with reweighted jumps based on similarity; 3) **IPFP** (Leordeanu et al., 2009) iteratively explores the optimal matching based on integer projection; 4) **Astar** (Riesen et al., 2007) finds the optimal matching between two graphs using priority search; 5) **NGM** (Wang et al., 2021a) uses graph convolution and Sinkhorn embedding network for learning the graph matching.

**Training Setup.** We train the models with a batch size of 16. The number of outer and inner iterations is set to 10 and 15 during training, and 20 and 25 during testing. For the Gumbel noise sampling in the QAP solver, we set the sampling count to 16 during training to improve efficiency, and fix it to 128 during testing. During training, we directly compute the unsupervised loss from the model output for backpropagation. During testing, to convert the output float similarity matrix into a strict 0-1 integer matching matrix, we use the Hungarian algorithm to obtain a solution that strictly satisfies the constraints.

**Evaluation.** We mainly use three metrics for evaluation. Let the model's output objective be $d$ and the objective provided in the dataset (optimal or sub-optimal) be $d^*$. 1) $\mathbf{obj} = \overline{d}$ means the average objective score $d$; 2) $\mathbf{gap} = \overline{d^* - d}$ represents the average gap between $d^*$ and $d$; 3) **Time**(sec/100it) is the average time (in seconds) taken to solve 100 instances.

**Results.** Performance across different methods on graph matching datasets is presented in Table 1. Almost all methods achieve objectives better than the ground truth, due to the scaling and noise perturbations added during dataset construction. Compared to other methods, BiQAP significantly outperforms in both objective quality and time efficiency. This demonstrates the effectiveness and efficiency of BiQAP in solving QAP problems.

Table 1: Graph matching test with varying scaling level $\delta_s$ and noise level $\sigma_n$. **GT** represents the objective given by the original ground truth matching, which is sub-optimal due to perturbations.

| ALGORITHM | GM-I ($\delta_s = 0.05, \sigma_n = 0.02$) | | | GM-II ($\delta_s = 0.3, \sigma_n = 0.2$) | | |
|---|---|---|---|---|---|---|
| | OBJ↑ | GAP↓ | TIME(s) | OBJ↑ | GAP↓ | TIME(s) |
| GT | 9216.74 | 0.00 | - | 8117.00 | 0.00 | - |
| SM | 9351.46 | -134.72 | 119.1 | 8277.45 | -160.45 | 124.1 |
| RRWM | 9275.63 | -58.89 | 1024.1 | 8147.77 | -30.78 | 1106.9 |
| IPFP | 9258.30 | -41.56 | 235.5 | 8171.04 | -54.04 | 268.6 |
| ASTAR | 9194.52 | 22.22 | 92355.7 | 8127.62 | -10.62 | 87693.4 |
| NGM | 9219.79 | -3.05 | 15.2 | 8145.27 | -28.27 | 26.8 |
| $\Delta$-SEARCH | 9670.62 | -453.88 | 82.7 | 8692.75 | -575.75 | 84.4 |
| BiQAP | **9708.35** | **-491.61** | **13.7** | **8724.73** | **-607.73** | **11.4** |

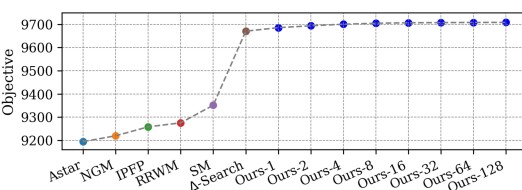

Figure 3: Sampling tests on GM-I dataset. The objective values are reported. Ours-$k$ indicates sampling with Gumbel noise of size $k$, where Ours-128 represents the sampling size we used.

**Sampling Tests.** To evaluate the solution quality of BiQAP, we conduct sampling tests to analyze the impact of different Gumbel sizes on performance. The visual results on GM-I dataset are shown in Fig. 3. Even with a Gumbel size of 1, BiQAP achieves an objective higher than all other methods. As the Gumbel size increases, the objective improves slightly, but the gains are minimal. This shows that the high quality of the solutions is primarily due to the effectiveness of BiQAP's design, rather than an increased sampling size. Additionally, experiments are conducted on other datasets, and the detailed results are provided in Appendix C.

---

[1]Methods $1 \sim 4$ are implemented by using Pygmtools (Wang et al., 2024).

### 4.2 CASE STUDY II: LARGE RANDOM DATASET

**Experiment Setting.** To better evaluate the capability of BiQAP, we construct an extremely large random dataset, generated by sampling from a uniform distribution $U(-2, 2)$ to create the matrices $\mathbf{F}_1$, $\mathbf{F}_2$, and $\mathbf{K}_p$. We sample three configurations: L500, L750 and L1000, where L500 indicates that the matrix sizes are $n_1 = n_2 = 500$. The problem sizes in this dataset are significantly larger than those in typical datasets, where problem sizes generally lower than 150. Methods based on LLQAP require computing the affinity matrix $\mathbf{K}$. However, when the matrix size $n$ ($n = n_1 = n_2$) is 500, 750 and 1000, respectively, the memory required for $\mathbf{K}$ amounts to 232.8GB, 1178.7GB and 3725.3GB. This makes LLQAP methods impractical for such large-scale QAP problems due to their excessive memory requirements. Thus, we only compare our model with the KBQAP method $\Delta$-Search. The evaluation metrics are the same as Sec. 4.1.

**Results.** From Table 2, our BiQAP surpasses $\Delta$-Search in both objective score and inference time. As the problem size of the dataset increases, the gap between BiQAP and $\Delta$-Search widens, indicating

Table 2: Performance comparison on large random datasets.

| ALGORITHM | L500 | | L750 | | L1000 | |
|---|---|---|---|---|---|---|
| | OBJ↑ | TIME(S) | OBJ↑ | TIME(S) | OBJ↑ | TIME(S) |
| $\Delta$-SEARCH | 29783.6 | 981 | 43120.7 | 2331 | 54307.7 | 4902 |
| BIQAP | **33167.1** | **210** | **60785.4** | **462** | **91613.7** | **850** |

that our BiQAP exhibits stronger problem-solving capabilities for larger datasets. Furthermore, BiQAP is significantly more time-efficient than $\Delta$-Search when handling large-scale problems, which is attributed to the efficient design of FormulaNet and our differentiable QAP solver.

### 4.3 CASE STUDY III: GRAPH EDIT DISTANCE

**Background and Preliminaries.** Computing the graph edit distance (GED) (Abu-Aisheh et al., 2015) is a widely used similarity measure for graphs and is known to be NP-hard. GED is defined as the minimum number of edit operations — adding/removing nodes/edges and change node labels — needed to transform one graph $G_1$ into another graph $G_2$. The GED problem can be reformulated as a KBQAP problem as shown in Eq. 8, which involves finding a matching matrix that represents the node alignment between the two graphs (the edit path). Detailed explanation and proof of this transformation are provided in Appendix F. Once the edit path is obtained, the GED can be easily computed. We use three real GED datasets for evaluation: AIDS, Linux (both with graphs of up to 10 nodes), and IMDB (with graphs of up to 89 nodes). The AIDS dataset contains various node labels, while the other two datasets lack node labels. Before evaluation, we preprocess these datasets to convert them into the KBQAP format and verify the correctness of Eq. 8 based on the ground truth.

$$- \text{GED}(G_1, G_2) = \max_{\mathbf{X}} J(\mathbf{X}), J(\mathbf{X}) = \text{tr}(\mathbf{X}^\mathsf{T} \mathbf{F}_1 \mathbf{X} \mathbf{F}_2) + \text{tr}(\mathbf{K}_p^\mathsf{T} \mathbf{X}) - n^2/4,$$
$$\text{s.t. } \mathbf{X} \in \{0,1\}^{n \times n}, \mathbf{X}\mathbf{1}_n = \mathbf{1}_n, \mathbf{X}^\mathsf{T}\mathbf{1}_n = \mathbf{1}_n, n = \max(n_1, n_2) = n_2 \tag{8}$$

Recent efforts in deep graph similarity learning (Bai et al., 2019; Bai & Zhao, 2021; Bai et al., 2020) use graph neural networks (Kipf & Welling, 2016; Scarselli et al., 2008) to directly regress graph similarity scores without explicitly incorporating the intrinsic combinatorial nature of GED, and thus fail to recover the edit path. As a result, the values predicted by these methods are often infeasible and of limited practical use. In contrast, the graph edit path (the optimization variable $\mathbf{X}$) is often of central interest in many applications (Dijkman et al., 2009; Fürstenau & Lapata, 2009; Chen et al., 2020), and most GED works (Neuhaus et al., 2006; Abu-Aisheh et al., 2015; Yang & Zou, 2021; Wang et al., 2021b; Piao et al., 2023) still focus on finding the edit path itself. Therefore, we aim to solve the edit path and compare our methods with other edit path-based approaches.

**Baselines.** In addition to the QAP baselines mentioned in Section 4.1, we compare our methods with approaches designed for finding the graph edit path: 1) **BeamSearch** (Neuhaus et al., 2006), an A*-beam search algorithm for GED; 2) **DF-GED** (Abu-Aisheh et al., 2015), an exact depth-first search method (limited to 200 seconds per instance); 3) **Noah** (Yang & Zou, 2021), an A*-beam search supervised by a GNN; 4) **Greedy**, using optimized settings for the Hungarian algorithm (Kuhn, 1955) and VJ algorithm (Jonker & Volgenant, 1988); 5) **GEDGNN** (Piao et al., 2023), the state-of-the-art method for graph edit path search using a $k$-best framework, also supervised by a GNN.

**Evaluations.** As in the previous experiments, we use the gap and Time(sec/100it) as evaluation metrics. Additionally, since the GED dataset includes ground truth, we introduce **acc** to represent the fraction of cases where $d \geq d^*$. In other words, **acc** measures the proportion of instances where the objective score exceeds or matches the ground truth score provided by the dataset.

Table 3: Performance on graph edit distance datasets.

| ALGORITHM | AIDS | | | LINUX | | | IMDB | | |
|---|---|---|---|---|---|---|---|---|---|
| | GAP↓ | ACC(%)↑ | TIME(S) | GAP↓ | ACC(%)↑ | TIME(S) | GAP↓ | ACC(%)↑ | TIME(S) |
| SM | 10.492 | 1.16 | 0.87 | 5.778 | 5.28 | 0.45 | 36.447 | 49.20 | 5.11 |
| RRWM | 10.677 | 0.96 | 20.55 | 5.162 | 14.40 | 13.75 | 36.410 | 48.73 | 18.77 |
| IPFP | 9.962 | 2.80 | 7.72 | 5.984 | 6.48 | 2.79 | 36.175 | 49.92 | 6.24 |
| ASTAR | 9.744 | 2.65 | 208.01 | 4.111 | 24.92 | 36.72 | 35.751 | 49.47 | 18.44 |
| NGM | 2.859 | 13.27 | 42.15 | 1.383 | 45.58 | 25.37 | 22.047 | 64.60 | 28.62 |
| Δ-SEARCH | 2.021 | 25.89 | **0.88** | 0.554 | 75.14 | **0.42** | 5.887 | 75.27 | **1.08** |
| BEAMSEARCH | 2.714 | 16.21 | 4.66 | 1.520 | 45.26 | 2.94 | 9.030 | 68.70 | 62.92 |
| DF-GED | 1.796 | 31.64 | 130.59 | **0.048** | **97.93** | 37.55 | 30.826 | 61.04 | 285.51 |
| GREEDY | 8.524 | 1.79 | 1.08 | 4.677 | 10.03 | 0.90 | 13.917 | 62.44 | 2.18 |
| NOAH | 3.078 | 6.34 | 168.39 | 1.747 | 8.71 | 77.24 | 10.172 | 52.29 | 5409.66 |
| GEDGNN | 1.515 | 42.60 | 73.34 | 0.224 | 91.18 | 24.62 | 3.133 | 81.35 | 132.54 |
| BIQAP | **0.053** | **94.99** | 4.62 | 0.055 | 97.92 | 4.21 | **0.228** | **96.94** | 9.28 |

Figure 4: Normalized objective score (lower is better) of our proposed method compared to other QAP solvers. Failed instances are plotted at the top of the y-axis (greater than 4.0). The instances are first divided based on whether BiQAP outperforms Sinkhorn-JA (Kushinsky et al., 2019), then sorted by the normalized score of BiQAP. NGM-G5k indicates the use of 5k Gumbel noise. BiQAP outperforms Sinkhorn-JA on 102 out of 134 instances and is able to solve all 134 instances, whereas both NGM-G5k and Sinkhorn-JA fail on some instances.

**Results.** The experimental results on Graph Edit Distance datasets are presented in Table 3. As shown, QAP-based methods generally perform slightly worse than methods specifically designed for GED. For example, on the AIDS dataset, the gap for all QAP-based methods is above 2.0. This may be due to the increased difficulty when transforming the GED problem into the KBQAP form. However, despite being a KBQAP-based method, BiQAP performs exceptionally well, with gaps below 0.3 and accuracy above 94% on all three datasets, while also being highly time-efficient. BiQAP not only outperforms all QAP-based methods but also significantly surpasses methods specifically designed for GED. This shows the powerful capability of BiQAP in solving KBQAP problems.

**Model Generalization.** From Fig. 4, we can find that our model performs very well on the corresponding test set of its own training set. Furthermore, it generalizes effectively to unseen datasets, even those with varying problem sizes and distinct data characteristics. Notably, the performance of the model

Table 4: Generalization tests on graph edit distance dataset. Our model is trained on three datasets, as depicted in the leftmost column. For each training setup, we evaluate our model across all testing datasets listed in the top row.

| TEST / TRAIN | AIDS | | LINUX | | IMDB | |
|---|---|---|---|---|---|---|
| | GAP↓ | ACC(%)↑ | GAP↓ | ACC(%)↑ | GAP↓ | ACC(%)↑ |
| AIDS | 0.053 | 94.99 | 0.008 | 99.62 | 2.001 | 93.08 |
| LINUX | 0.201 | 84.15 | 0.055 | 97.92 | 0.294 | 98.54 |
| IMDB | 0.131 | 89.69 | 0.081 | 96.04 | 0.228 | 96.94 |

trained on the AIDS dataset and tested on Linux exceeds that of the model trained and tested on Linux itself (both with problem sizes smaller than 10). This may be due to the higher quality of the AIDS dataset and its inclusion of node labels, which likely enhance the model's ability to generalize across datasets with similar sizes but different characteristics. However, the performance drops significantly when trained on AIDS and tested on IMDB, due to the problem size of the IMDB dataset is much larger (up to 89), and the model trained on AIDS is unable to generalize well across datasets with distinct problem sizes. In contrast, models trained on the Linux and IMDB datasets show a more balanced generalization ability across both problem size and dataset characteristics.

## 4.4 CASE STUDY IV: QAPLIB

**Experiment Setting.** QAPLIB (Burkard et al., 1997) consists of 134 real-world QAP instances from 15 categories, including problems like hospital facility layout planning (Hahn & Krarup, 2001). These problems are formulated as KBQAP (Eq. 2), but with $\mathbf{K}_p$ being a zero matrix. Since the objective in QAPLIB is minimization, we negate the $\mathbf{F}_1$ matrix to align with our KBQAP formula-

Table 5: Best-performing occurrence count across different categories.

| CATEGORY | BUR | CHR | ELS | ESC | HAD | KRA | LIPA | NUG | ROU | SCR | SKO | STE | TAI | THO | WIL | TOTAL |
|---|---|---|---|---|---|---|---|---|---|---|---|---|---|---|---|---|
| #INSTANCES | 8 | 14 | 1 | 19 | 5 | 3 | 16 | 15 | 3 | 3 | 13 | 3 | 26 | 3 | 2 | 134 |
| SM | 0 | 0 | 0 | 1 | 0 | 0 | 0 | 0 | 0 | 0 | 0 | 0 | 0 | 0 | 0 | 1 |
| RRWM | 0 | 0 | 0 | 2 | 0 | 0 | 0 | 0 | 0 | 0 | 0 | 0 | 0 | 0 | 0 | 2 |
| SINKHORN-JA | 0 | 10 | 1 | 0 | 0 | 0 | 15 | 1 | 0 | 0 | 0 | 0 | 3 | 0 | 1 | 31 |
| NGM-G5K | 0 | 0 | 0 | 3 | 2 | 0 | 0 | 2 | 0 | 0 | 0 | 0 | 1 | 0 | 0 | 8 |
| BIQAP | 8 | 4 | 0 | 13 | 3 | 3 | 1 | 12 | 3 | 3 | 13 | 3 | 22 | 3 | 1 | 92 |

tion. Due to structural similarities in each category, we train one network per category. Our method is fairly compared with RRWM, SM, NGM, and Sinkhorn-JA (Kushinsky et al., 2019), a heuristic method designed for QAPLIB based on the QAP formulation. NGM reports results using 5k Gumbel noise. Given the large variation across instances in QAPLIB, we report the normalized score for each instance, which is computed using the upper bound provided by the dataset, and further normalized by the baseline solver, spectral matching (SM) (Leordeanu & Hebert, 2005):

$$norm\_score = \frac{solved\_score - upper\_bound}{SM\_score - upper\_bound} \quad (9)$$

**Results.** Detailed scores and timing results are available in Appendix H. The visualization for each instance is shown in Fig. 4. We observe that our method outperforms learning-free methods SM, RRWM, and the learning-based method NGM, while being comparable to and even superior to Sinkhorn-JA. It is important to note that due to the high complexity of LLQAP, NGM fails to solve the `tai256c` instance (requiring 275GB of GPU memory for intermediate computations). Additionally, for problem instances not reported in (Kushinsky et al., 2019), we assume Sinkhorn-JA fails to find any feasible solution, as the original paper provides no explanation for the missing instances. Compared to NGM, with our Gumbel size set to 128 versus NGM-G5k's Gumbel size

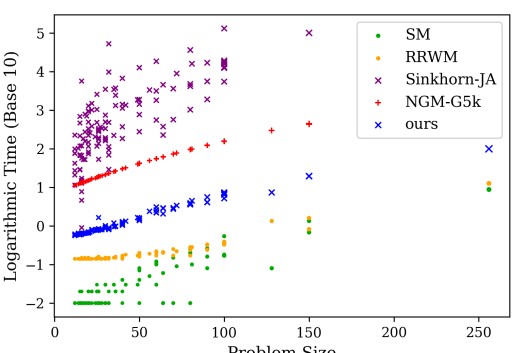

Figure 5: Inference time(sec) vs problem size $n$, with a base-10 log-scale y-axis. Each point represents the time to solve an instance.

of 5k, our method outperforms it on the majority of instances and successfully solves `tai256c`, which NGM fails to solve. Against Sinkhorn-JA, BiQAP outperforms it on 102 out of 134 instances.

Further evaluation is presented in Table 5 and Fig. 5. Our BiQAP finds the best solution in 92 out of 134 instances, while the learning-based NGM-G5k and learning-free Sinkhorn-JA outperform on 8 and 31 instances, respectively. This indicates that BiQAP can solve a wider range of problems compared to traditional solvers. More importantly, it performs inference much faster than both NGM and Sinkhorn-JA, achieving strong results in both solution quality and computational efficiency.

## 4.5 ADDITIONAL EXPERIMENTS

We conduct additional important experiments, detailed in the appendix. The case study on the Traveling Salesman Problem (TSP) (Appendix A) shows that our BiQAP is competitive with other QAP-based methods. The ablation study (Appendix B) strongly highlights the effectiveness of our FormulaNet and Gromov-Sinkhorn QAP solver, both of which are essential components. The sampling tests (Appendix C) indicates that the high quality of the solutions primarily results from the effectiveness of BiQAP's design, rather than the increased sampling size.

## 5 CONCLUSION

We have presented a time-efficient bi-level framework, BiQAP, to solve the Koopmans-Beckmann QAP problem. The outer level optimize the original objective, while the inner minimization leverages FormulaNet to learn a new QAP and solve it by the differentiable Gromov-Sinkhorn QAP solver capable of producing high-quality solutions. To the best of our knowledge, this is the first end-to-end QAP neural framework that does not heavily rely on random sampling techniques or complex search algorithms. Extensive experimental results across five tasks show its superiority in both effectiveness and efficiency compared to learning-free and learning-based methods.

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

## A  CASE STUDY V: TRAVELING SALESMAN PROBLEM

**Background and Preliminary.** The Traveling Salesman Problem (TSP) is a well-known problem in combinatorial optimization: given a set of cities and the distances between each pair of cities, the objective is to find the shortest possible route that visits each city exactly once and return the first city. TSP can be formulated as a permutation-based QAP (Goh et al., 2022), but this formulation is not consistent with KBQAP. We adopt a new formulation to represent TSP as a KBQAP. Specifically, the variable $\mathbf{X}_{v,j}$ is an indicator of whether city $v$ is the $j$-th city to be visited. Here, $\mathbf{F}_1 \in \mathbb{R}^{n \times n}$ represents an indicator matrix, $\mathbf{F}_2 \in \mathbb{R}^{n \times n}$ is the distance matrix, and $\mathbf{K}_p$ is a zero matrix, where $n = n_1 = n_2$ is the number of cities. The construction of $\mathbf{F}_1$ is shown in Eq. 10. $\mathbf{F}_{2;u,v}$ denotes the distance between city $u$ and city $v$. Detailed proof of this formulation is provided in Appendix G.

$$\mathbf{F}_{1;i,j} = \left\{ \begin{array}{ll} -0.5, & \text{if } |i-j| = 1 \text{ or } |i-j| = n-1 \\ 0, & \text{otherwise.} \end{array} \right. \tag{10}$$

**Dataset and Baselines.** In this experiment, we focus on 2D Euclidean TSP. For each instance, we generate 50 points from $U(0,1) \times U(0,1)$ (TSP-50), where $U$ denotes a uniform distribution. We use the Concorde TSP solver (David et al., 2006) to obtain ground-truth optimal solutions. A total of 128K instances are sampled for training, and 1.28K instances for testing. The evaluation metrics include objective (obj), gap, and Time (sec/100it), where gap is the difference between the optimal objective and the obtained objective. Similar to previous experiments, we compare our method with QAP-based methods such as SM, RRWM, IPFP, Astar, NGM, and $\Delta$-Search.

Notably, for many tasks, obtaining exact ground-truth solutions can be extremely difficult, or even infeasible, such as the Graph Matching in Sec. 4.1. Therefore, we employ an unsupervised loss, which does not rely on ground-truth data, making our method widely applicable. However, since Concorde provides optimal solutions for TSP-50, we use these solutions as ground truth. To further evaluate the effectiveness of our framework, we utilize a variant, "BiQAP-s," where we replace the unsupervised loss with the supervised Binary Cross-Entropy (BCE) loss. This allows us to assess the performance of our framework in a supervised setting as well.

**Results.** Performance on the TSP-50 is shown in Table 6. Compared to the Concorde solver, which is specifically designed for solving TSP, the gap for QAP-based methods is relatively large. This is because the KBQAP formulation of the TSP problem is more complex. During the transformation process, some crucial information may be obscured. This may increases the likelihood of local optima, making the problem more difficult to solve. In other words, TSP may be easy for classical heuristics designed for routing problems but becomes more challenging when translated into the QAP formulation. It is worth noting that SM, RRWM, IPFP, and Astar exhibit low and

Table 6: Results on TSP-50. "BiQAP-s" refers to BiQAP trained by supervised loss.

| ALGORITHM | OBJ↑ | GAP↓ | TIME(S) |
|---|---|---|---|
| CONCORDE | -5.69 | 0.00 | 5.13 |
| SM | -26.06 | 20.38 | 7730.5 |
| RRWM | -26.03 | 20.34 | 8543.6 |
| IPFP | -26.03 | 20.34 | 105.5 |
| ASTAR | -26.05 | 20.36 | 22274.6 |
| NGM | -21.02 | 15.34 | 36.0 |
| $\Delta$-SEARCH | -9.01 | 3.32 | 27.2 |
| BIQAP-S | **-7.38** | **1.70** | **10.4** |
| BIQAP | -8.53 | 2.84 | 11.5 |

similar objective values, likely because these QAP-based methods struggle to capture the problem characteristics in the KBQAP format for this dataset. However, despite this complexity, both BiQAP and BiQAP-s outperform other QAP-based methods in terms of both effectiveness and efficiency. Notably, BiQAP-s delivers superior performance, indicating that for the TSP task, using supervised data often leads to better results.

## B  ABLATION STUDY

We validate the effectiveness of FormulaNet and our QAP solver in Table 7 on both graph matching datasets and large random datasets. We compare several settings of BiQAP with $\Delta$-Search, exploring FormulaNet's ability to learn a new instance and the impact of the number of outer/inner iterations of our QAP solver on performance.

From the results of "BiQAP w/o FN," it is evident that removing FormulaNet significantly degrades overall performance, with the objective score notably lower than both $\Delta$-Search and other BiQAP settings that retain FormulaNet, especially on the L500, L750 and L1000 datasets. This suggests

Table 7: Ablation study on graph matching datasets (GM-I and GM-II) and large random datasets (L500, L750 and L1000). "BiQAP w/o FN" denotes our method without FormulaNet, where our QAP solver takes the original instance as input. "BiQAP-10/15" indicates that the number of outer and inner iterations of our QAP solver during the inference stage is 10 and 15, respectively. Similarly, "BiQAP-20/25" and "BiQAP-30/35" follow the same notation, with "BiQAP-20/25" being the setting used in our main experiments.

| ALGORITHM | GM-I | | GM-II | | L500 | | L750 | | L1000 | |
|---|---|---|---|---|---|---|---|---|---|---|
| | OBJ↑ | TIME(S) | OBJ↑ | TIME(S) | OBJ↑ | TIME(S) | OBJ↑ | TIME(S) | OBJ↑ | TIME(S) |
| Δ-SEARCH | 9670.6 | 82.7 | 8692.8 | 84.4 | 29783.6 | 981.0 | 43120.7 | 2331.0 | 54307.7 | 4902.2 |
| BIQAP W/O FN | 9396.4 | 12.1 | 8357.1 | 10.9 | 3048.8 | 182.6 | 4375.3 | 372.4 | 5784.4 | 781.1 |
| BIQAP-10/15 | **9708.5** | 9.3 | 8724.5 | 11.2 | 31429.6 | 183.9 | 58745.9 | 394.2 | 89809.0 | 763.9 |
| BIQAP-20/25(OURS) | 9708.4 | 13.7 | 8724.7 | 11.4 | **33167.1** | 210.0 | **60785.4** | 462.4 | **91613.7** | 850.2 |
| BIQAP-30/35 | 9707.7 | 18.0 | **8724.8** | 20.7 | 32068.3 | 282.6 | 57547.3 | 625.6 | 86379.4 | 1204.2 |

that FormulaNet effectively learns a simpler new instance for the QAP solver, enabling the model to achieve better results.

Regarding the number of outer/inner iterations of the QAP solver, "BiQAP-10/15" performs best on the GM-I dataset and "BiQAP-30/35" yields the best results on the GM-II dataset. However, the differences between these iteration settings are not substantial, and all configurations outperform Δ-Search. This indicates that, for graph matching datasets, the number of outer/inner iterations has a limited impact on model performance within a reasonable range, highlighting the robustness of our method to hyperparameter variations. On the L500, L750, and L1000 datasets, "BiQAP-20/25" achieves the best performance, with other iteration settings performing slightly worse but still outperforming Δ-Search.

## C  EXPERIMENTS ON SAMPLING SIZE

To better explore the effect of sampling size on model performance, we conduct the sample size study on the GM-I dataset from Graph Matching, the AIDS dataset from GED, and the L500 dataset from the Large Random Datasets. Furthermore, we compare the results with some baseline methods. Table 8 shows the experimental results for these three datasets. From the experimental results,

Table 8: Sampling size tests on three datasets against prominent baselines. GEDGNN (Piao et al., 2023) is tailored for the AIDS dataset, while NGM (Wang et al., 2021a) cannot handle large instances such as L500 due to modeling constraints. "Ours-k" denotes our model with a sample size of k.

| METHOD | GM-I OBJ↑ | L500 OBJ↑ | AIDS GAP↓ | AIDS ACC(%)↑ |
|---|---|---|---|---|
| GEDGNN | - | - | 1.515 | 42.6 |
| NGM | 9219.79 | - | 2.859 | 13.27 |
| Δ-SEARCH | 9670.62 | 29783.6 | 2.021 | 25.89 |
| OURS-1 | 9685.68 | 29036.4 | 1.214 | 46.76 |
| OURS-2 | 9693.83 | 29945.1 | 0.897 | 58.14 |
| OURS-4 | 9700.50 | 30781.7 | 0.489 | 73.39 |
| OURS-8 | 9704.76 | 31261.0 | 0.275 | 80.91 |
| OURS-16 | 9706.79 | 31826.5 | 0.137 | 87.84 |
| OURS-32 | 9707.54 | 32349.2 | 0.087 | 92.17 |
| OURS-64 | 9708.14 | 32848.6 | 0.065 | 93.84 |
| OURS-128 | 9708.35 | 33167.1 | 0.053 | 94.99 |

we observe that as the sample size increases, the model's performance improves. However, when the sample size reaches around 64 to 128, the improvement becomes less significant, indicating diminishing returns.

Moreover, compared to the AIDS dataset, the performance improvements from increasing the sample size are less pronounced on the GM-I and L500 datasets. We believe that this is because the instances in the GED dataset are much smaller than those in the Graph Matching and Large Random datasets. As a result, increasing the sample size allows for better exploration of the solution space, making it easier to find the optimal solution. For larger datasets, although increasing the sample size explores a larger portion of the solution space, the search space is so vast that the performance gains are not as noticeable. It is also worth noting that even with a sample size of 1, our method still

shows significant performance advantages over the baselines across all three datasets. Only on the L500 dataset does the objective with sample size 1 slightly lag behind $\Delta$-Search, but as the sample size increases, our model surpasses it.

In conclusion, the increase in the number of samples does have an impact on performance, which depends on the characteristics of the dataset. But even with a sample size of 1, our model consistently outperforms other baselines. This indicates that the high quality of the solutions primarily results from the effectiveness of BiQAP's design, rather than the increased sampling size.

## D    DETAILED PROOF OF EQ. 6

Here we give a simple proof of Eq. 6. We use Lagrangian multipliers:

$$\mathcal{L} = -\text{tr}(\mathbf{X}^\mathsf{T}\mathbf{F}_1\mathbf{X}\mathbf{F}_2) - \text{tr}(\mathbf{K}_p^\mathsf{T}\mathbf{X}) - \epsilon H(\mathbf{X}) - \langle \alpha, \mathbf{X}\mathbf{1} - \mathbf{1} \rangle - \langle \beta, \mathbf{X}^\mathsf{T}\mathbf{1} - \mathbf{1} \rangle \tag{11}$$

Next, we need to find the stationary points of the Lagrangian multiplier. We take first-order derivatives to $\mathcal{L}$:

$$\begin{aligned}
\frac{\partial\mathcal{L}}{\partial\mathbf{X}} &= -\frac{\partial\text{tr}(\mathbf{X}^\mathsf{T}\mathbf{F}_1\mathbf{X}\mathbf{F}_2)}{\partial\mathbf{X}} - \frac{\partial\text{tr}(\mathbf{K}_p^\mathsf{T}\mathbf{X})}{\partial\mathbf{X}} - \epsilon\frac{\partial H(\mathbf{X})}{\partial\mathbf{X}} - \alpha - \beta \\
&= -\mathbf{F}_1\mathbf{X}\mathbf{F}_2 - \mathbf{F}_1^\mathsf{T}\mathbf{X}\mathbf{F}_2^\mathsf{T} - \mathbf{K}_p + \epsilon\log\mathbf{X} - \alpha - \beta = 0
\end{aligned} \tag{12}$$

We further simplify the equation of Lagrangian into the following term:

$$\mathbf{X} = \text{diag}(e^{\alpha/\epsilon})e^{\left(\mathbf{F}_1\mathbf{X}\mathbf{F}_2 + \mathbf{F}_1^\mathsf{T}\mathbf{X}\mathbf{F}_2^\mathsf{T} + \mathbf{K}_p\right)/\epsilon}\text{diag}(e^{\beta/\epsilon}) \tag{13}$$

Therefore, we can directly interpret the problem as an optimal transport problem, with the cost matrices computed using $\mathbf{X}$. Through an iterative process, we calculate $\mathbf{X}$ as follows:

$$\mathbf{X} = \text{Sinkhorn}(\mathbf{C}, \epsilon), \text{ where } \mathbf{C} = -\mathbf{F}_1\mathbf{X}\mathbf{F}_2 - \mathbf{F}_1^\mathsf{T}\mathbf{X}\mathbf{F}_2^\mathsf{T} - \mathbf{K}_p. \tag{14}$$

## E    FORMULANET

The block design of the Mamba-based FormulaNet is illustrated in Fig. 6. Given an input matrix $M \in \mathbb{R}^{n_1 \times n_2}$, we first flatten the matrix into a vector and use a projection layer to map this vector into a $d$-dimensional space, resulting in a sequence of size $n_1 n_2 \times d$.

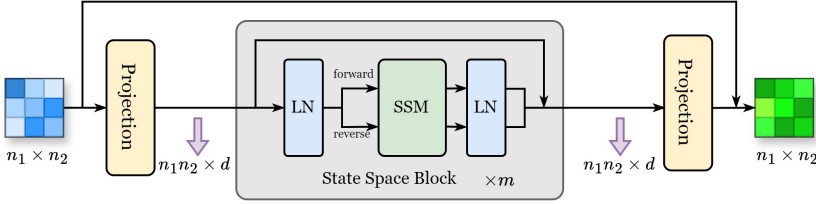

Figure 6: Architecture of our proposed Mamba-based FormulaNet. Various structures can be employed as FormulaNet, as long as they are capable of accepting arbitrary shapes as input, and the matrix size of both the input and output is $n_1 \times n_2$.

Then, we utilize a series of State Space Blocks. Within each State Space Block, to prevent training instability, the input sequence passes through a layer normalization (LN) layer at the beginning and the end of the block. Sometimes, using only one layer normalization yields better results, depending on the characteristics of the dataset. In the middle, we employ a State Space Model (SSM) based on the Mamba architecture to capture the long-term spatial dependencies. Since the input sequence does not have a strict order, we feed it into the SSM in both forward and reverse directions. The SSM follows the structure proposed in (Gu & Dao, 2023). After passing through the final layer normalization, the outputs of the sequences from both directions are merged, and a residual connection is applied to produce the output of the State Space Block.

After processing through a series of State Space Blocks, the sequence is fed into a final projection layer, which maps the sequence back from the $d$-dimensional space to a 1-dimensional vector, restoring the original matrix shape of $n_1 \times n_2$. Finally, we add a residual connection to obtain the output.

## F  KBQAP FORMULA OF GRAPH EDIT DISTANCE

### F.1  DEFINITION AND EXPLANATION

Here is a brief explanation of Eq. 8. Consider two given graphs $G_1$ and $G_2$. Without loss of generality, assume:

$$n_1 = |G_1|, \quad n_2 = |G_2|, \quad n_1 \le n_2, \quad n = \max(n_1, n_2) = n_2. \tag{15}$$

Let $\mathbf{A}_1 \in \mathbb{R}^{n_1 \times n_1}$ and $\mathbf{A}_2 \in \mathbb{R}^{n_2 \times n_2}$ denote the adjacency matrices of the two graphs, and $\mathbf{L}_1 \in \mathbb{Z}^{n_1 \times d}$ and $\mathbf{L}_2 \in \mathbb{Z}^{n_2 \times d}$ denote the node label matrices, where each row is a one-hot vector of dimension $d$. We introduce a padding operation $\text{pad}(\mathbf{A}, n, k)$, which pads a matrix $\mathbf{A}$ (where $\mathbf{A} \in \mathbb{R}^{m_1 \times m_2}$, $m_1 \le n$, $m_2 \le n$) to size $n \times n$, filling the padded elements with a value $k$.

From adjacency matrices $\mathbf{A}_1$ and $\mathbf{A}_2$, we define $\hat{\mathbf{F}}_1 \in \mathbb{R}^{n_1 \times n_1}$ and $\hat{\mathbf{F}}_2 \in \mathbb{R}^{n_2 \times n_2}$ as follows:

$$\hat{\mathbf{F}}_{i,j} = \begin{cases} -0.5, & \text{if } \mathbf{A}_{i,j} = 0 \text{ or } i = j \\ 0.5, & \text{otherwise.} \end{cases} \tag{16}$$

For the KBQAP formulation, we have:

$$\mathbf{F}_1 = \text{pad}(\hat{\mathbf{F}}_1, n, -0.5) \in \mathbb{R}^{n \times n}, \tag{17}$$

$$\mathbf{F}_2 = \hat{\mathbf{F}}_2 \in \mathbb{R}^{n \times n}, \tag{18}$$

$$\mathbf{K}_p = \text{pad}(\mathbf{L}_1 \mathbf{L}_2^\mathsf{T} - \mathbf{1}^{n_1 \times n_2}, n_2, -1) \in \mathbb{R}^{n \times n}. \tag{19}$$

Given that $\text{GED}(G_1, G_2)$ represents the graph edit distance between $G_1$ and $G_2$, the KBQAP formulation is:

$$-\,\text{GED}(G_1, G_2) = \max_{\mathbf{X}} J(\mathbf{X}), \quad J(\mathbf{X}) = \text{tr}(\mathbf{X}^\mathsf{T} \mathbf{F}_1 \mathbf{X} \mathbf{F}_2) + \text{tr}(\mathbf{K}_p^\mathsf{T} \mathbf{X}) - \frac{n^2}{4},$$

$$\text{s.t. } \mathbf{X} \in \{0,1\}^{n \times n}, \quad \mathbf{X}\mathbf{1}_n = \mathbf{1}_n, \quad \mathbf{X}^\mathsf{T}\mathbf{1}_n = \mathbf{1}_n, \quad n = \max(n_1, n_2) = n_2. \tag{20}$$

### F.2  PROOF

In our KBQAP formulation for the graph edit distance problem (Eq. 20), the three matrices $\mathbf{F}_1$, $\mathbf{F}_2$, and $\mathbf{K}_p$ are all of size $n \times n$, with $\mathbf{F}_1$ and $\mathbf{F}_2$ being symmetric matrices. Therefore, using the properties of the trace of matrices, we have:

$$\begin{aligned} J(\mathbf{X}) &= \text{tr}(\mathbf{X}^\mathsf{T} \mathbf{F}_1 \mathbf{X} \mathbf{F}_2) + \text{tr}(\mathbf{K}_p^\mathsf{T} \mathbf{X}) \\ &= \langle \mathbf{F}_1 \mathbf{X} \mathbf{F}_2, \mathbf{X} \rangle_F + \langle \mathbf{K}_p, \mathbf{X} \rangle_F \\ &= \sum_{i,j} \mathbf{F}_{1;i,j} \mathbf{F}_{2;h(j),h(i)} + \sum_i \mathbf{K}_{p;i,h(i)} \end{aligned} \tag{21}$$

where $\langle \cdot, \cdot \rangle$ denotes the Frobenius inner product, and $h(i) = \arg\max \mathbf{X}_i$ is the index of the column where the $i$-th row of $\mathbf{X}$ has a value of 1. From Eq. 21 and Eqs. 17, 18, and 19, we can derive:

$$\sum_{i,j} \mathbf{F}_{1;i,j} \mathbf{F}_{2;h(j),h(i)} = \sum_{i,j} \mathbf{F}_{1;i,j} \mathbf{F}_{2;h(i),h(j)}$$

$$= 2 \sum_{i<j} \mathbf{F}_{1;i,j} \mathbf{F}_{2;h(i),h(j)} + \sum_{i=1}^{n} \mathbf{F}_{1;i,i} \mathbf{F}_{2;h(i),h(i)} \tag{22}$$

$$= 2 \sum_{i<j} \mathbf{F}_{1;i,j} \mathbf{F}_{2;h(i),h(j)} + \frac{n}{4}$$

Given the definition of the function $h(i)$, the corresponding matching matrix $\mathbf{X}$ is determined, which gives the node matching permutation between $G_1$ and $G_2$. Since $n_1 \leq n_2$, we add $n_2 - n_1$ isolated nodes to $G_1$ to obtain $G_1'$, so that both $G_1'$ and $G_2$ are graphs with $n$ nodes ($n = n_2$). For a permutation function $h$, let $e_{1;i,j}$ represent the edge between nodes $i$ and $j$ in $G_1'$, and $e_{2;h(i),h(j)}$ represent the corresponding edge between nodes $h(i)$ and $h(j)$ in $G_2$. We define condition function:

$$\delta_C = \begin{cases} 1, & \text{if condition } C \text{ is satisfied} \\ 0, & \text{if condition } C \text{ is not satisfied} \end{cases} \tag{23}$$

And we define these conditions:

$$C_{i,j}^1 : \text{both } e_{1;i,j} \text{ and } e_{2;h(i),h(j)} \text{ either exist or do not exist}$$

$$C_{i,j}^2 : \text{exactly one of } e_{1;i,j} \text{ or } e_{2;h(i),h(j)} \text{ exists}$$

Using Eqs. 16, 17, and 18, we obtain:

$$2 \sum_{i<j} \mathbf{F}_{1;i,j} \mathbf{F}_{2;h(i),h(j)} = \frac{1}{2} \sum_{i<j} (\delta_{C_{i,j}^1} - \delta_{C_{i,j}^2})$$

$$= \frac{1}{2} \sum_{i<j} (\delta_{C_{i,j}^1} - \delta_{C_{i,j}^2} - 1) + \frac{1}{4} n(n-1) = \sum_{i<j} (-\delta_{C_{i,j}^2}) + \frac{1}{4} n(n-1) \tag{24}$$

Therefore, according to Eqs. 22 and 24, the first term of $J(\mathbf{X})$ in the KBQAP formula Eq. 20 is:

$$\text{tr}(\mathbf{X}^\mathsf{T} \mathbf{F}_1 \mathbf{X} \mathbf{F}_2) = 2 \sum_{i<j} \mathbf{F}_{1;i,j} \mathbf{F}_{2;h(i),h(j)} + \frac{n}{4} = \sum_{i<j} (-\delta_{C_{i,j}^2}) + \frac{n^2}{4} \tag{25}$$

Next, given that each row of the node label matrices $\mathbf{L}_1$ and $\mathbf{L}_2$ is a one-hot vector, we define the conditions:

$$C_{i,j}^3 : \text{the label of node } i \text{ in } G_1 \text{ matches the label of node } j \text{ in } G_2$$

$$C_{i,j}^4 : \text{the label of node } i \text{ in } G_1' \text{ does not match the label of node } j \text{ in } G_2$$

Thus we have:

$$(\mathbf{L}_1 \mathbf{L}_2^\mathsf{T})_{i,j} = \delta_{C_{i,j}^3} = \begin{cases} 1, & \text{if } \mathbf{L}_{1;i} = \mathbf{L}_{2;j} \\ 0, & \text{otherwise} \end{cases} \tag{26}$$

From the previous content, we know that $G_1'$ is formed by adding $n_2 - n_1$ isolated nodes to $G_1$. Clearly, when computing the graph edit distance, the number of these newly added nodes needs to be accounted for in the distance. To keep the distance unchanged, instead of explicitly considering the distance for adding these $n_2 - n_1$ nodes, we use a change in node labels as a substitute for adding these nodes. Therefore, the labels of these newly added nodes must not match any existing node labels in the original graph. Based on Eq. 19, we can further express $\mathbf{K}_p$ as follows:

$$\mathbf{K}_{p;i,j} = \begin{cases} \delta_{C_{i,j}^3} - 1, & \text{if } i \leq n_1 \text{ and } j \leq n_2 \\ -1, & \text{if } n_1 < i \leq n_2 \end{cases} = -\delta_{C_{i,j}^4}. \tag{27}$$

According to Eqs. 21 and 27, the second term of $J(\mathbf{X})$ in the KBQAP formula Eq. 20 is:

$$\text{tr}(\mathbf{K}_p^\mathsf{T} \mathbf{X}) = \sum_i \mathbf{K}_{p;i,h(i)} = \sum_i (-\delta_{C_{i,h(i)}^4}) \tag{28}$$

Thus, KBQAP formula Eq. 20 is:

$$J(\mathbf{X}) = \text{tr}(\mathbf{X}^\mathsf{T} \mathbf{F}_1 \mathbf{X} \mathbf{F}_2) + \text{tr}(\mathbf{K}_p^\mathsf{T} \mathbf{X}) - \frac{n^2}{4}$$

$$= \sum_{i<j} (-\delta_{C_{i,j}^2}) + \frac{n^2}{4} + \sum_i (-\delta_{C_{i,h(i)}^4}) - \frac{n^2}{4} \tag{29}$$

$$= -\left( \sum_{i<j} \delta_{C_{i,j}^2} + \sum_i \delta_{C_{i,h(i)}^4} \right)$$

Given a permutation function $h$, the distance computed using this permutation is denoted as $GED_h(G_1, G_2)$. In Eq. 29, $\sum_{i<j} \delta_{C^2_{i,j}}$ represents the number of added or removed edges for the given permutation function $h$, while $\sum_i \delta_{C^4_{i,h(i)}}$ denotes the number of changed labels (including the $n_2 - n_1$ nodes added to $G_1$). Therefore, we have:

$$J(\mathbf{X}) = -\left( \sum_{i<j} \delta_{C^2_{i,j}} + \sum_i \delta_{C^4_{i,h(i)}} \right) = -GED_h(G_1, G_2) \tag{30}$$

Thus, we have:

$$\max_{\mathbf{X}} J(\mathbf{X}) \iff \max_h -GED_h(G_1, G_2) \iff \min_h GED_h(G_1, G_2) \iff GED(G_1, G_2) \tag{31}$$

## G  KBQAP FORMULA OF TRAVELING SALESMAN PROBLEM

Here we provide a proof for the construction of the KBQAP-formulated TSP instance as discussed in section A. The matrix $\mathbf{F}_1 \in \mathbb{R}^{n \times n}$ is constructed according to Eq. 32. The matrix $\mathbf{F}_2 \in \mathbb{R}^{n \times n}$ represents the distance matrix, where $\mathbf{F}_{2;u,v}$ denotes the distance between city $u$ and city $v$. The matrix $\mathbf{K}_p$ is a zero matrix. This formulation enables us to express the TSP as a KBQAP instance (Eq. 2).

$$\mathbf{F}_{1;i,j} = \begin{cases} -0.5, & \text{if } |i - j| = 1 \text{ or } |i - j| = n - 1 \\ 0, & \text{otherwise.} \end{cases} \tag{32}$$

From Eq. 21, using the definitions of $\mathbf{F}_1$, $\mathbf{F}_2$, and $\mathbf{K}_p$, we have:

$$J(\mathbf{X}) = \sum_{i,j} \mathbf{F}_{1;i,j} \mathbf{F}_{2;h(j),h(i)} + \sum_i \mathbf{K}_{p;i,h(i)} = \sum_{i,j} \mathbf{F}_{1;i,j} \mathbf{F}_{2;h(i),h(j)}$$

$$= 2 \sum_{i<j} \mathbf{F}_{1;i,j} \mathbf{F}_{2;h(i),h(j)} + \sum_{i=1}^n \mathbf{F}_{1;i,i} \mathbf{F}_{2;h(i),h(i)} = 2 \sum_{i<j} \mathbf{F}_{1;i,j} \mathbf{F}_{2;h(i),h(j)} \tag{33}$$

$$= 2 \sum_{i=1}^n -0.5 \times \mathbf{F}_{2;h(i),h(\mathrm{mod}(i,n)+1)} = -\sum_{i=1}^n d_{h(i),h(\mathrm{mod}(i,n)+1)}$$

where $d_{u,v}$ represents the distance between city $u$ and city $v$. Note that $\mathbf{F}_1$ and $\mathbf{F}_2$ are symmetric matrices. In $\sum_{i=1}^n d_{h(i),h(\mathrm{mod}(i,n)+1)}$, $h(1)$ denotes the first city, $h(i)$ represents the $i$-th city, and $d_{h(n),h(\mathrm{mod}(n,n)+1)} = d_{h(n),h(1)}$ corresponds to the distance from the $n$-th city back to the first city. Therefore, $J(\mathbf{X}) = \sum_{i=1}^n d_{h(i),h(\mathrm{mod}(i,n)+1)}$ represents the total distance of visiting the cities in the order specified by the rows of $\mathbf{X}$ and returning to the starting city. Let $TSP(h)$ represent the route distance based on the permutation function $h$. Thus, we have:

$$\max_{\mathbf{X}} J(\mathbf{X}) \iff \max_h -TSP(h) \iff \min_h TSP(h) \tag{34}$$

# H DETAILED PER-INSTANCE RESULTS ON QAPLIB

Table 9: Detailed per-instance results of solved scores and inference times on the QAPLIB dataset. A '-' indicates that the instance could not be solved by the method.

| instance | Score | | | | | | Time(Sec) | | | | |
|---|---|---|---|---|---|---|---|---|---|---|---|
| | Upper | SM | RRWM | Sinkhorn-JA | NGM-G5k | Ours | SM | RRWM | Sinkhorn-JA | NGM-G5k | Ours |
| bur26a | 5426670 | 6533340 | 6663181 | 5688893 | 5621774 | 5547613 | 0.02 | 0.15 | 309.90 | 19.34 | 1.64 |
| bur26b | 3817852 | 4690772 | 4741283 | 4053243 | 3927943 | 3910002 | 0.01 | 0.15 | 191.70 | 19.21 | 0.85 |
| bur26c | 5426795 | 6537412 | 6474996 | 5639665 | 5608065 | 5571621 | 0.01 | 0.15 | 136.90 | 18.74 | 0.78 |
| bur26d | 3821225 | 4649645 | 4678974 | 3985052 | 3962317 | 3924012 | 0.02 | 0.16 | 276.60 | 18.78 | 0.81 |
| bur26e | 5386879 | 6711029 | 6619788 | 5539241 | 5536142 | 5522832 | 0.01 | 0.16 | 52.90 | 18.62 | 0.80 |
| bur26f | 3782044 | 4723824 | 4814298 | 3979071 | 3949711 | 3885136 | 0.01 | 0.15 | 173.60 | 18.59 | 0.80 |
| bur26g | 10117172 | 12168111 | 12336830 | 10624776 | 10433439 | 10361552 | 0.01 | 0.15 | 292.80 | 18.55 | 0.78 |
| bur26h | 7098658 | 8753694 | 8772077 | 7453329 | 7348866 | 7287865 | 0.01 | 0.15 | 330.40 | 18.64 | 0.78 |
| chr12a | 9552 | 50732 | 43624 | 9552 | 14940 | 13332 | 0.01 | 0.14 | 75.70 | 11.26 | 0.56 |
| chr12b | 9742 | 46386 | 73860 | 9742 | 14984 | 13502 | 0.01 | 0.14 | 75.10 | 11.34 | 0.57 |
| chr12c | 11156 | 57404 | 50130 | 11156 | 16346 | 13690 | 0.01 | 0.14 | 97.90 | 11.20 | 0.56 |
| chr15a | 9896 | 77094 | 90870 | 11616 | 20442 | 16006 | 0.01 | 0.14 | 683.60 | 12.60 | 0.59 |
| chr15b | 7990 | 77430 | 115556 | 7990 | 22048 | 20354 | 0.01 | 0.14 | 461.90 | 12.61 | 0.59 |
| chr15c | 9504 | 64198 | 70738 | 9504 | 24190 | 16410 | 0.01 | 0.14 | 214.10 | 12.59 | 0.59 |
| chr18a | 11098 | 94806 | 115328 | 11948 | 33124 | 23986 | 0.01 | 0.14 | 781.50 | 14.00 | 0.63 |
| chr18b | 1534 | 4054 | 3852 | 2690 | 2504 | 1622 | 0.01 | 0.14 | 52.10 | 13.97 | 0.64 |
| chr20a | 2192 | 11154 | 13970 | 4624 | 5178 | 4504 | 0.02 | 0.14 | 1285.80 | 15.02 | 0.70 |
| chr20b | 2298 | 9664 | 14168 | 3400 | 5766 | 3682 | 0.02 | 0.14 | 911.30 | 14.96 | 0.66 |
| chr20c | 14142 | 112406 | 195572 | 40464 | 49770 | 39276 | 0.02 | 0.14 | 945.00 | 14.93 | 0.70 |
| chr22a | 6156 | 16732 | 15892 | 9258 | 9348 | 8726 | 0.02 | 0.14 | 1488.40 | 16.88 | 0.76 |
| chr22b | 6194 | 13294 | 13658 | 6634 | 9006 | 8416 | 0.02 | 0.14 | 1005.30 | 16.10 | 0.73 |
| chr25a | 3796 | 21526 | 32060 | 5152 | 11648 | 10478 | 0.03 | 0.15 | 2553.20 | 17.93 | 0.77 |
| els19 | 17212548 | 33807116 | 74662642 | 18041490 | 27029748 | 24558642 | 0.01 | 0.14 | 700.00 | 14.49 | 0.65 |
| esc16a | 68 | 98 | 80 | 100 | 78 | 68 | 0.00 | 0.14 | 12.80 | 13.00 | 0.65 |
| esc16b | 292 | 318 | 294 | 304 | 292 | 292 | 0.00 | 0.14 | 4.60 | 12.97 | 0.65 |
| esc16c | 160 | 276 | 204 | 266 | 174 | 160 | 0.01 | 0.14 | 7.70 | 13.00 | 0.66 |
| esc16d | 16 | 48 | 44 | 58 | 20 | 18 | 0.01 | 0.14 | 14.60 | 12.95 | 0.66 |
| esc16e | 28 | 52 | 50 | 44 | 32 | 28 | 0.00 | 0.14 | 13.50 | 13.00 | 0.63 |
| esc16f | 0 | 0 | 0 | 0 | 0 | 0 | 0.00 | 0.14 | 0.90 | 12.98 | 0.64 |
| esc16g | 26 | 44 | 52 | 52 | 32 | 26 | 0.00 | 0.14 | 17.10 | 13.04 | 0.61 |
| esc16h | 996 | 1292 | 1002 | 1282 | 1004 | 996 | 0.00 | 0.14 | 15.10 | 12.95 | 0.60 |
| esc16i | 14 | 54 | 28 | 36 | 18 | 14 | 0.02 | 0.14 | 5625.60 | 12.97 | 0.62 |
| esc16j | 8 | 22 | 18 | 18 | 8 | 8 | 0.01 | 0.14 | 13.00 | 13.04 | 0.61 |
| esc32a | 130 | 426 | 240 | 456 | 298 | 182 | 0.01 | 0.15 | 91.80 | 22.91 | 0.82 |
| esc32b | 168 | 460 | 400 | 416 | 368 | 188 | 0.00 | 0.15 | 28.90 | 22.87 | 0.80 |
| esc32c | 642 | 770 | 650 | 886 | 754 | 642 | 0.00 | 0.15 | 112.40 | 22.95 | 0.85 |
| esc32d | 200 | 360 | 224 | 356 | 284 | 218 | 0.00 | 0.14 | 68.40 | 22.84 | 0.88 |
| esc32e | 2 | 68 | 6 | 46 | 2 | 2 | 0.02 | 0.15 | 9661.40 | 22.88 | 0.85 |
| esc32g | 6 | 36 | 10 | 46 | 10 | 6 | 0.01 | 0.15 | 52135.20 | 22.82 | 0.85 |
| esc32h | 438 | 602 | 506 | - | 534 | 454 | 0.00 | 0.15 | - | 22.79 | 0.81 |
| esc64a | 116 | 254 | 124 | 276 | 200 | 134 | 0.01 | 0.20 | 225.80 | 61.55 | 2.06 |
| esc128 | 64 | 202 | 78 | - | 242 | 192 | 0.08 | 1.34 | - | 297.84 | 7.34 |
| had12 | 1652 | 1894 | 2090 | - | 1700 | 1722 | 0.01 | 0.14 | - | 11.28 | 0.62 |
| had14 | 2724 | 3310 | 3494 | 2916 | 2866 | 2782 | 0.01 | 0.14 | 102.20 | 12.13 | 0.60 |
| had16 | 3720 | 4390 | 4646 | 3978 | 3902 | 3826 | 0.01 | 0.14 | 56.70 | 12.88 | 0.63 |
| had18 | 5358 | 6172 | 6540 | 5736 | 5558 | 5558 | 0.01 | 0.15 | 271.40 | 13.90 | 0.65 |
| had20 | 6922 | 8154 | 8550 | 7464 | 7300 | 7204 | 0.01 | 0.14 | 328.40 | 14.97 | 0.68 |
| kra30a | 88900 | 148690 | 136830 | 125290 | 114410 | 110490 | 0.01 | 0.14 | 491.60 | 21.36 | 1.01 |
| kra30b | 91420 | 150760 | 141550 | 126980 | 118130 | 111240 | 0.01 | 0.14 | 489.90 | 21.35 | 1.00 |
| kra32 | 88700 | 145310 | 148730 | 128120 | 120930 | 112370 | 0.01 | 0.15 | 479.60 | 22.96 | 0.90 |
| lipa20a | 3683 | 3956 | 3940 | 3683 | 3853 | 3824 | 0.01 | 0.14 | 271.10 | 14.89 | 0.68 |
| lipa20b | 27076 | 36502 | 38236 | 27076 | 33125 | 32408 | 0.01 | 0.14 | 73.30 | 15.04 | 0.67 |
| lipa30a | 13178 | 13861 | 13786 | 13178 | 13631 | 13576 | 0.01 | 0.15 | 191.90 | 21.18 | 0.86 |
| lipa30b | 151426 | 198434 | 201775 | 151426 | 187607 | 185425 | 0.03 | 0.15 | 160.50 | 21.35 | 0.86 |
| lipa40a | 31538 | 32736 | 32686 | 31538 | 32454 | 32381 | 0.01 | 0.14 | 183.20 | 30.09 | 1.04 |
| lipa40b | 476581 | 628272 | 647295 | 476581 | 601848 | 596653 | 0.04 | 0.17 | 369.30 | 30.10 | 1.07 |
| lipa50a | 62093 | 64070 | 64162 | 62642 | 63671 | 63531 | 0.01 | 0.16 | 275.20 | 41.04 | 1.38 |
| lipa50b | 1210244 | 1589128 | 1591109 | 1210244 | 1523856 | 1512221 | 0.08 | 0.20 | 763.50 | 41.25 | 1.39 |
| lipa60a | 107218 | 109861 | 110468 | 108456 | 109595 | 109445 | 0.03 | 0.17 | 551.50 | 54.70 | 2.22 |
| lipa60b | 2520135 | 3303961 | 3300291 | 2520135 | 3208501 | 3187208 | 0.10 | 0.22 | 1796.20 | 55.37 | 2.19 |
| lipa70a | 169755 | 173649 | 173569 | 172504 | 173220 | 172948 | 0.01 | 0.17 | 565.80 | 72.61 | 2.73 |
| lipa70b | 4603200 | 6055613 | 6063182 | 4603200 | 5890161 | 5860517 | 0.15 | 0.25 | 3592.80 | 72.90 | 2.93 |

Table 10: Continued detailed per-instance results of solved scores and inference times on the QAPLIB dataset. A '-' indicates that the instance could not be solved by the method.

| instance | Score | | | | | | Time(Sec) | | | | |
|---|---|---|---|---|---|---|---|---|---|---|---|
| | Upper | SM | RRWM | Sinkhorn-JA | NGM-G5k | Ours | SM | RRWM | Sinkhorn-JA | NGM-G5k | Ours |
| lipa80a | 253195 | 258345 | 258608 | 257395 | 257663 | 257524 | 0.01 | 0.17 | 1023.40 | 94.93 | 3.66 |
| lipa80b | 7763962 | 10231797 | 10223697 | 7763962 | 9983040 | 9957201 | 0.20 | 0.27 | 4158.00 | 95.20 | 3.71 |
| lipa90a | 360630 | 367384 | 367370 | 366649 | 366508 | 366295 | 0.08 | 0.24 | 1889.50 | 122.14 | 4.06 |
| lipa90b | 12490441 | 16291267 | 16514577 | 12490441 | 16076956 | 16027722 | 0.26 | 0.33 | 5544.50 | 122.29 | 4.09 |
| nug12 | 578 | 886 | 1038 | 682 | 634 | 626 | 0.01 | 0.14 | 11.40 | 11.29 | 0.61 |
| nug14 | 1014 | 1450 | 1720 | - | 1156 | 1088 | 0.01 | 0.14 | - | 12.10 | 0.65 |
| nug15 | 1150 | 1668 | 2004 | 1448 | 1318 | 1238 | 0.01 | 0.14 | 69.60 | 12.56 | 0.63 |
| nug16a | 1610 | 2224 | 2626 | 1940 | 1836 | 1768 | 0.01 | 0.14 | 118.70 | 13.01 | 0.61 |
| nug16b | 1240 | 1862 | 2192 | 1492 | 1396 | 1386 | 0.01 | 0.15 | 66.80 | 12.96 | 0.63 |
| nug17 | 1732 | 2452 | 2934 | 2010 | 1980 | 1892 | 0.01 | 0.14 | 181.60 | 13.44 | 0.65 |
| nug18 | 1930 | 2688 | 3188 | 2192 | 2242 | 2136 | 0.01 | 0.15 | 155.20 | 13.88 | 0.65 |
| nug20 | 2570 | 3450 | 4174 | 3254 | 2936 | 2816 | 0.01 | 0.15 | 146.70 | 14.87 | 0.67 |
| nug21 | 2438 | 3702 | 4228 | 3064 | 2916 | 2732 | 0.01 | 0.14 | 256.80 | 15.52 | 0.69 |
| nug22 | 3596 | 5896 | 6382 | 3988 | 4298 | 4098 | 0.01 | 0.14 | 382.60 | 16.09 | 0.81 |
| nug24 | 3488 | 4928 | 5720 | 4424 | 4234 | 3952 | 0.01 | 0.14 | 202.60 | 17.34 | 0.77 |
| nug25 | 3744 | 5332 | 5712 | 4302 | 4420 | 4266 | 0.01 | 0.14 | 478.70 | 17.95 | 0.81 |
| nug27 | 5234 | 7802 | 8626 | 6244 | 6208 | 6262 | 0.01 | 0.14 | 360.30 | 19.34 | 0.80 |
| nug28 | 5166 | 7418 | 8324 | 6298 | 6128 | 6140 | 0.01 | 0.14 | 339.60 | 19.96 | 0.79 |
| nug30 | 6124 | 8956 | 10034 | 7242 | 7294 | 7006 | 0.01 | 0.14 | 330.70 | 21.28 | 0.89 |
| rou12 | 235528 | 325404 | 377168 | 276446 | 264898 | 246942 | 0.01 | 0.14 | 41.90 | 11.35 | 0.56 |
| rou15 | 354210 | 489350 | 546526 | 390810 | 403872 | 386744 | 0.01 | 0.14 | 66.20 | 12.49 | 0.60 |
| rou20 | 725522 | 950018 | 1010554 | 823298 | 817776 | 810398 | 0.01 | 0.14 | 115.10 | 14.97 | 0.66 |
| scr12 | 31410 | 71392 | 95134 | 45334 | 36292 | 35896 | 0.01 | 0.14 | 20.80 | 11.33 | 0.58 |
| scr15 | 51140 | 104308 | 101714 | 74632 | 68768 | 61910 | 0.02 | 0.14 | 117.10 | 12.62 | 0.59 |
| scr20 | 110030 | 263058 | 350528 | 171260 | 154636 | 145130 | 0.01 | 0.14 | 220.80 | 14.96 | 0.65 |
| sko42 | 15812 | 20770 | 23612 | 19058 | 18716 | 18220 | 0.03 | 0.18 | 1342.50 | 32.01 | 1.33 |
| sko49 | 23386 | 29616 | 34548 | 27160 | 27554 | 26726 | 0.03 | 0.17 | 1849.20 | 39.80 | 1.64 |
| sko56 | 34458 | 44594 | 49650 | 40954 | 40684 | 39668 | 0.05 | 0.19 | 3318.10 | 48.94 | 2.38 |
| sko64 | 48498 | 60878 | 65540 | 55738 | 56222 | 55016 | 0.06 | 0.21 | 4533.60 | 61.82 | 2.90 |
| sko72 | 66256 | 82156 | 89264 | 76332 | 76870 | 75490 | 0.09 | 0.22 | 8845.20 | 77.14 | 3.37 |
| sko81 | 90998 | 112838 | 118372 | 105246 | 104710 | 102670 | 0.10 | 0.28 | 15863.80 | 97.28 | 4.46 |
| sko90 | 115534 | 140840 | 148784 | 133818 | 132942 | 131066 | 0.16 | 0.32 | 16796.60 | 122.47 | 5.56 |
| sko100a | 152002 | 185738 | 184854 | 176626 | 172810 | 170726 | 0.18 | 0.33 | 18370.80 | 155.41 | 6.68 |
| sko100b | 153890 | 185366 | 189502 | 177398 | 175588 | 173428 | 0.17 | 0.33 | 15432.10 | 155.32 | 6.74 |
| sko100c | 147862 | 178710 | 188756 | 169566 | 169806 | 167492 | 0.17 | 0.38 | 13000.40 | 155.69 | 7.13 |
| sko100d | 149576 | 181328 | 186086 | 170648 | 170816 | 168410 | 0.17 | 0.33 | 17350.90 | 155.58 | 7.07 |
| sko100e | 149150 | 180062 | 192342 | 171656 | 170958 | 168652 | 0.17 | 0.37 | 16240.40 | 155.24 | 6.81 |
| sko100f | 149036 | 177518 | 189284 | 171296 | 169986 | 167710 | 0.17 | 0.37 | 19155.60 | 155.17 | 7.41 |
| ste36a | 9526 | 30030 | 33294 | 17938 | 16768 | 11602 | 0.02 | 0.15 | 2415.20 | 26.21 | 0.97 |
| ste36b | 15852 | 176526 | 193046 | 47616 | 43248 | 25474 | 0.02 | 0.16 | 3718.00 | 26.32 | 0.98 |
| ste36c | 8239110 | 24530792 | 28908062 | 14212212 | 12988352 | 9683098 | 0.02 | 0.15 | 1312.10 | 26.42 | 0.98 |
| tai12a | 224416 | 318032 | 392004 | 245012 | 255158 | 254566 | 0.01 | 0.14 | 27.10 | 11.38 | 0.56 |
| tai12b | 39464925 | 96190153 | 124497790 | 81727424 | 47252044 | 45642400 | 0.01 | 0.14 | 225.10 | 11.35 | 0.56 |
| tai15a | 388214 | 514304 | 571952 | 471272 | 436968 | 426198 | 0.01 | 0.14 | 28.20 | 12.51 | 0.58 |
| tai15b | 51765268 | 702925159 | 702292926 | 52585356 | 52871608 | 52441320 | 0.01 | 0.14 | 29.00 | 12.56 | 0.59 |
| tai17a | 491812 | 669712 | 738566 | 598716 | 544754 | 543196 | 0.01 | 0.14 | 52.40 | 13.94 | 0.63 |
| tai20a | 703482 | 976236 | 1012228 | 849082 | 806382 | 787724 | 0.01 | 0.14 | 82.60 | 14.91 | 0.65 |
| tai20b | 122455319 | 394836310 | 602903767 | 220470588 | 140704160 | 157404704 | 0.02 | 0.15 | 489.90 | 14.89 | 0.69 |
| tai25a | 1167256 | 1485502 | 1536172 | 1341104 | 1352912 | 1314338 | 0.02 | 0.14 | 116.00 | 18.03 | 0.76 |
| tai25b | 344355646 | 764920942 | 1253946482 | 798113083 | 518647040 | 495104384 | 0.02 | 0.14 | 1040.00 | 17.95 | 0.77 |
| tai30a | 1818146 | 2210304 | 2305048 | 2072218 | 2065706 | 2045994 | 0.03 | 0.15 | 175.30 | 21.35 | 0.85 |
| tai30b | 637117113 | 1008164383 | 1766978330 | 1114514832 | 896379008 | 862257600 | 0.03 | 0.15 | 3464.20 | 21.32 | 0.90 |
| tai35a | 2422002 | 3030184 | 3100748 | 2820060 | 2786748 | 2755974 | 0.03 | 0.15 | 221.10 | 25.35 | 0.95 |
| tai35b | 283315445 | 454981851 | 574511546 | 446783959 | 377687744 | 357131136 | 0.03 | 0.15 | 3440.60 | 25.36 | 1.19 |
| tai40a | 3139370 | 3825396 | 3985684 | 3547918 | 3610604 | 3559256 | 0.04 | 0.16 | 1121.60 | 30.20 | 1.14 |
| tai40b | 637250948 | 1165811212 | 1423772477 | 1019672934 | 917498816 | 831085824 | 0.04 | 0.15 | 6646.70 | 29.92 | 1.24 |
| tai50a | 4938796 | 6078426 | 6203546 | 5569952 | 5677282 | 5633704 | 0.07 | 0.19 | 1418.50 | 41.38 | 1.52 |
| tai50b | 458821517 | 796553600 | 790688128 | 696556852 | 614638528 | 574294144 | 0.08 | 0.18 | 12552.00 | 41.11 | 1.55 |
| tai60a | 7205962 | 8614998 | 8731620 | 8243624 | 8281996 | 8192368 | 0.11 | 0.21 | 3121.10 | 55.35 | 3.07 |
| tai60b | 608215054 | 1089964672 | 1279537664 | 978843717 | 862969152 | 801760000 | 0.12 | 0.20 | 18385.70 | 55.34 | 2.57 |
| tai64c | 1855928 | 5893540 | 6363888 | 3189566 | 2133738 | 1986866 | 0.01 | 0.21 | 373.40 | 61.70 | 2.84 |
| tai80a | 13499184 | 15665790 | 16069786 | 15352662 | 15283138 | 15141412 | 0.20 | 0.28 | 4745.20 | 95.09 | 3.55 |
| tai80b | 818415043 | 1338090880 | 1410723456 | 1215586531 | 1120577408 | 1048145664 | 0.25 | 0.24 | 35995.40 | 94.92 | 3.91 |
| tai100a | 21052466 | 24176962 | 24446982 | 23787764 | 23644528 | 23526762 | 0.34 | 0.39 | 5447.50 | 156.06 | 5.16 |
| tai100b | 1185996137 | 1990209280 | 2192130048 | 1589275900 | 1612020992 | 1571683072 | 0.54 | 0.34 | 130312.50 | 156.20 | 6.41 |
| tai150b | 498896643 | 662657408 | 755505920 | - | 628349568 | 601300864 | 1.35 | 1.60 | - | 433.41 | 19.51 |
| tai256c | 44759294 | 77548512 | 77161352 | - | - | 49431412 | 8.76 | 12.61 | - | - | 99.14 |
| tho30 | 149936 | 230828 | 267194 | 202844 | 185622 | 181272 | 0.01 | 0.14 | 739.10 | 21.38 | 0.91 |
| tho40 | 240516 | 375154 | 440146 | 314070 | 304878 | 295214 | 0.02 | 0.15 | 1407.00 | 30.18 | 1.29 |
| tho150 | 8133398 | 10000616 | 10689758 | 9508422 | 9557766 | 9455292 | 0.68 | 0.82 | 99778.20 | 443.22 | 19.39 |
| wil50 | 48816 | 56588 | 60420 | 54030 | 53418 | 52632 | 0.04 | 0.18 | 1867.00 | 42.11 | 1.60 |
| wil100 | 273038 | 305030 | 307258 | 292118 | 294172 | 292308 | 0.17 | 0.34 | 12315.50 | 154.60 | 7.28 |

