# OpenReview forum: "BiQAP: Neural Bi-level Optimization-based Framework for Solving Quadratic Assignment Problems"
_ICLR.cc/2025/Conference — Submitted to ICLR 2025_

### Official Review · Reviewer_WMpS · 2024-10-30

**Soundness:** 4
**Presentation:** 4
**Contribution:** 2
**Rating:** 6
**Confidence:** 3

**Summary:**

This paper presents an unsupervised learning approach to solve QAPs. The proposed method includes two main components: a "FormulaNet" that generates a new QAP and a differentiable solver utilizing the Sinkhorn algorithm to solve this generated QAP. Experimental results show certain advantages over the baseline methods.

**Strengths:**

1. The paper is well-written, with a logical flow.
2. Unlike existing graph-matching methods, which use optimal transportation as the implicit optimization problem, the proposed approach aims to solve a new QAP. This is novel.
3. Extensive experiments are conducted, covering different types of QAPs and various exsting baseline methods.

**Weaknesses:**

I don't see any major weaknesses, but a few improvements could enhance this paper:

1. The implicit optimization problem appears crucial, as described by the author in lines 280-285. Including a baseline where FormulaNet produces an optimal transportation problem could provide a more robust validation and enhance readers' understanding.
2. The author should clarify the choice of randomly generated datasets over publicly available ones, as the latter might be more convincing.

**Questions:**

See weaknesses.

---

> ### Author Response · Authors · 2024-11-18
> **Answer (1/1, 1 in total)**
>
> >***Q1: "The implicit optimization problem appears crucial, as described by the author in lines 280-285 (290-295 in revised version). Including a baseline where FormulaNet produces an optimal transportation problem could provide a more robust validation and enhance readers' understanding."***
>
> A1. Thank you for your reminder. We will discuss the implicit optimization further in our paper, which actually serves as the inner optimization in our framework. In fact, as discussed in [1,2], many layers (e.g., Softmax, tanh, Sigmoid, Sinkhorn) can be understood as solving operators for their implicit optimization, and many papers have attempted to improve learning by designing and solving these implicit optimizations. (We provide an example with Softmax in the related work section of the updated version) When the layer is used as the final network layer for output, the entire learning process can be viewed as solving the implicit optimization based on the features derived from the neural network. From this perspective, the work of NGM [3] using Sinkhorn activation essentially solves an entropically regularized transport problem:
>
> $$
> \min_{\mathbf{X1}=\mathbf{1},\mathbf{X^\top1}=\mathbf{1}}  \langle  \mathbf{C}^\theta,\mathbf{X} \rangle- \epsilon H(\mathbf{X}) \qquad (a)
> $$
>
> where $\mathbf{C}^\theta$ represents the cosine distance learned from the neural network. In contrast, we employ a more complex implicit optimization as the inner optimization:
>
> $$
> \min_{\mathbf{X}} -\text{tr}(\mathbf{X}^\top\mathbf{F}_1^{\theta}\mathbf{X}\mathbf{F}_2^{\theta})-\text{tr}({\mathbf{K}_p^{\theta}}^\top\mathbf{X})-\epsilon H(\mathbf{X}), \qquad (b)
> $$
>
> $$
> \text{ s.t. }\mathbf{X1}=\mathbf{1},\mathbf{X}^{\top}\mathbf{1}\leq\mathbf{1}
> $$
> Since the QAP problem we are solving (see Eq. 2) typically has a non-convex objective, the convex optimization objective in Eq. (a) may be too simple, limiting its ability to fit Eq. 2, thus resulting in suboptimal performance. In this paper, instead of the entropic optimal transportation objective, we employ a more complex non-convex optimization, i.e., the entropic QAP, which better approximates the objective of QAP. Of course, using other more complex optimization problems to approximate QAP or other combinatorial optimization objectives could yield better results, which is an area of further research for us.
>
> [1] Regularized Optimal Transport Layers for Generalized Global Pooling Operations. TPAMI, 2023.
>
> [2] Implicit deep learning. SIAM Journal on Mathematics of Data Science 3.3 (2021): 930-958.
>
> [3] Neural Graph Matching Network: Learning Lawler's Quadratic Assignment Problem with Extension to Hypergraph and Multiple-graph Matching. TPAMI, 2021.
>
> >***Q2: "The author should clarify the choice of randomly generated datasets over publicly available ones, as the latter might be more convincing."***
>
> A2. Yes, we should indeed clarify the origins of the datasets we use, specifying whether they are publicly available real datasets, publicly available generated datasets, or datasets we generated ourselves:
>
> - The graph matching datasets mentioned in [2] are publicly available; however, as explained in lines 63–72 of our paper, "the ground-truth matching in supervised data may not necessarily be the optimal solution in the context of QAP optimization." Therefore, we cannot use visual graph matching datasets, which may contain errors for QAP optimization. Following the approach in [1] and [2], we conduct QAP-based graph matching experiments. While these papers provide methods for generating QAP-based graph matching datasets, they do not release the datasets. As a result, we followed the methods in [1] and [2] to generate our own graph matching datasets.
>
> - The Graph Edit Distance (GED) datasets are publicly available, real datasets that have been used in previous works such as [3] and [4].
>
> - The Large Random Datasets (L500/750/1000) are generated datasets containing extremely large instances, created specifically to evaluate the capability of our model. To our knowledge, no publicly available datasets of this size currently exist, so we generate them ourselves.
>
> - QAPLIB ([2], [5]) is a publicly available real dataset.
>
> - The Traveling Salesman Problem (TSP) datasets are publicly available generated datasets, and we use the same generation code in [1] and [6].
>
>
> [1] Towards Quantum Machine Learning for Constrained Combinatorial Optimization: a Quantum QAP Solver. ICML, 2023.
>
> [2] Neural Graph Matching Network: Learning Lawler's Quadratic Assignment Problem with Extension to Hypergraph and Multiple-graph Matching. TPAMI, 2021.
>
> [3] Computing Graph Edit Distance via Neural Graph Matching. VLDB, 2023.
>
> [4] Noah: Neural-optimized A* Search Algorithm for Graph Edit Distance Computation. ICDE, 2021.
>
> [5] QAPLIB - A Quadratic Assignment Problem Library - Problem instances and solutions.
>
> [6] The Transformer Network for the Traveling Salesman Problem.

---

> ### Author Response · Authors · 2024-11-24
> **Friendly Reminder for Feedback Before Discussion Deadline**
>
> **Dear Reviewer,**
>
> Thank you for your valuable response. As the discussion deadline approaches, we kindly request your feedback of our response to the concerns you previously raised. We are also open to any further questions or suggestions that you may have.
>
> We sincerely appreciate your time and attention. Your guidance is truly invaluable to our research.
>
> Best regards,
>
> The Authors

---

> > ### Comment · Reviewer_WMpS · 2024-11-25
> >
> > Thanks for your responses. I will remain the score.

---

### Official Review · Reviewer_L8oe · 2024-11-02

**Soundness:** 3
**Presentation:** 2
**Contribution:** 4
**Rating:** 6
**Confidence:** 3

**Summary:**

This paper proposes BiQAP, a procedure for solving Koopmans-Beckmann Quadratic Assignment Problems. The method formulates a bi-level optimization procedure, where the inner problem solves an entropy-regularized QAP. The problem data of the inner problem is predicted from the original problem data using a sequence to sequence neural network that is invariant to the size of the involved matrices. This neural network can be trained in an unsupervised manner, which removes the need for access to expensive gold solutions. The inner problem is solved with a differentiable approximate Gromov-Wasserstein Sinkhorn solver.
The method is tested on a wide set of experimental setups, including synthetically generated graph matching instances and more realistic graph edit distance (formulated as QAP) and QAP instances. Throughout the extensive evaluation, BiQAP produces strong results both in terms of achieved objective values of the computed solution as well as the required computation time, outperforming all of the other compared methods.

**Strengths:**

- The paper has a thorough experimental evaluation with strong results, and substantially advances the state of the art in multiple experimental setups.
- The experimental setup and evaluation are well-described and easy to follow.
- The approach taken (bi-level problem with entropy-regularized inner problem on predicted parameters) seems more widely applicable and could serve as a new paradigm in solving difficult non-convex and combinatorial problems.

**Weaknesses:**

- The paper lacks theoretical analysis of the bi-level optimization problem formulation (3).
- In the first part of the paper, I found the language used often imprecise and a bit confusing. Especially figure 1 seems misleading, because using QAP for visual keypoint matching is not what is done in this work, instead machine learning is used to improve the QAP solving itself. The presentation could be improved here.
- The ablation study on the number of samples in Fig 3 shows that in this setting the results are not very sensitive to this hyperparameter. This hyperparameter will be important for a practitioner so this ablation study should be repeated on a different experiment where it potentially makes a difference, e.g. on the GED experiment.
- The regularization strength $\epsilon$ of the inner problem is most likely an important hyperparameter (affecting the inner solution and its differentiation), it should be discussed and experimentally tested. What happens when it is set to very large or very small values?
- No ablation for different architectures of the FormulaNet is included. It would be important to see how the SSM compares to GNN or Transformer-based architectures.

**Questions:**

- What is the theoretical justification for using Gumbel noise to sample the initial $X^{(0)}$?
- I assume the time measurements in the tables and plots are given in seconds? Unless I missed it, this information should be added in.

---

> ### Author Response · Authors · 2024-11-18
> **Answer (1/3, 3 in total)**
>
> >***Q1: "What is the theoretical justification for using Gumbel noise to sample the initial $X^{(0)}$?"***
>
> A1. The choice of sampling method has little impact on the relationships in our QAP solver. Our goal is to ensure that various initializations of the latent space consistently move toward minimizing the loss. Inspired by the application of Gumbel sampling in the Sinkhorn algorithm (Gumbel Sinkhorn, [1]), we adopt Gumbel sampling within our framework. Notably, the first outer iteration in our QAP solver can be viewed as a Gumbel Sinkhorn step, which motivates our choice of Gumbel sampling.
>
> Additionally, we have tested alternative initialization methods, such as Gaussian sampling and uniform sampling. The experimental results on Graph Matching dataset GM-I are as follows:
>
> | Sampling method | GM-I Obj$\uparrow$ |
> |---|:---:|
> | Gaussian | 9704.56 |
> | Uniform | 9709.92 |
> | Gumbel | 9708.35 |
>
> We observe that these sampling strategies have almost no significant effect on the final performance of the model. Thus, we select gumbel sampling as the initialization method.
>
>
> [1] Learning Latent Permutations with Gumbel-Sinkhorn Networks. ICLR 2018.
>
> >***Q2: "I assume the time measurements in the tables and plots are given in seconds? Unless I missed it, this information should be added in."***
>
> A2. Yes, the time measurements are indeed in seconds. This is clarified in Section 4.1, "Evaluation" paragraph, where we state: "Time(sec/100it) is the average time taken to solve 100 instances." This explanation indicates that all subsequent time measurements represent the time required to solve 100 instances in seconds.
>
> However, as this clarification appears only once, readers might overlook it. To address this, we will include additional explanations in the revised version to ensure clarity and prevent any misunderstanding.
>
> >***Q3: "The paper lacks theoretical analysis of the bi-level optimization problem formulation (3) (Eq. 4 in revised version)."***
>
> A3. Thank you for your suggestion. We have further expanded the discussion on bi-level optimization in the updated version of our paper. Specifically, **our bi-level optimization is essentially a loss design used for learning QAP**. In the inner optimization, we first input the original QAP parameters $\mathbf{F}_1, \mathbf{F}_2,$ and $\mathbf{K}_p$ into the neural network, to obtain a new entropic regularized QAP with parameters $\mathbf{F}_1^{\theta}, \mathbf{F}_2^{\theta},$ and $\mathbf{K}_p^{\theta}$ ：
>
> $$
> \min_{\mathbf{X}} -\text{tr}(\mathbf{X}^\top\mathbf{F}_1^{\theta}\mathbf{X}\mathbf{F}_2^{\theta})-\text{tr}({\mathbf{K}_p^{\theta}}^\top\mathbf{X})-\epsilon H(\mathbf{X}),
> $$
>
> $$
> \text{ s.t. }\mathbf{X1}=\mathbf{1},\mathbf{X}^{\top}\mathbf{1}\leq\mathbf{1}
> $$
> This optimization is the inner optimization in Eq. 3 (Eq. 4 in revised version), where the Gromov-Sinkhorn algorithm (see Algorithm 1) is applied as a differentiable solver to solve the new entropic regularized QAP and obtain the solution $\mathbf{X}^\theta$. Note that the differentiable solver acts as an activation layer similar to softmax or Sinkhorn, allowing gradient backpropagation. Finally, given the calculated solution $\mathbf{X}^\theta$, we minimize the negative objective of the original QAP in the outer optimization, which serves as the loss function:
> $$
> \min_\theta  -\text{tr}\big(({\mathbf{X}^{\theta}})^\top\mathbf{F}_1\mathbf{X}^{\theta}\mathbf{F}_2\big)-\text{tr}(\mathbf{K}_p^\top\mathbf{X}^{\theta})
> $$
> Certainly, the real learning process involves the random sampling of the inner optimization, which we will not discuss further. For more specific details, please refer to Section 3.2.
>
> >***Q4: "Especially figure 1 seems misleading, because using QAP for visual keypoint matching is not what is done in this work, instead machine learning is used to improve the QAP solving itself. The presentation could be improved here."***
>
> A4. No, Figure 1 is crucial as it highlights the distinction between our work and previous graph matching approaches ([1], [2], [3]), while also clarifying why we do not directly compare our method with these works. Through an example, Figure 1 demonstrates that the ground truth matching in graph matching studies may result in a lower objective value, whereas visually incorrect matches obtained via Gurobi can achieve a higher objective value. As a result, visual graph matching differs significantly from solving QAP in combinatorial optimization.
>
> For a detailed explanation, please refer to lines 63–72 of our paper.
>
> [1] Graph-Context Attention Networks for Size-Varied Deep Graph Matching. CVPR, 2022.
>
> [2] Deep Learning of Partial Graph Matching via Differentiable Top-K. CVPR, 2023.
>
> [3] Learning deep graph matching with channel-independent embedding and Hungarian attention. ICLR, 2020.

---

> > ### Author Response · Authors · 2024-11-18
> > **Answer (2/3, 3 in total)**
> >
> > >***Q5: "The ablation study on the number of samples in Fig 3 shows that in this setting the results are not very sensitive to this hyperparameter. This hyperparameter will be important for a practitioner so this ablation study should be repeated on a different experiment where it potentially makes a difference, e.g. on the GED experiment."***
> >
> > A5. Thank you for your suggestion! The study of sample size should be extended to other experiments. Due to space limitations, we conduct the study only on the GM dataset, but we have added supplementary experiments here. It is important to note that, in addition to Figure 3, we also perform other ablation studies in Appendix B.
> >
> > We conduct the sample size study on the GM-I dataset from Graph Matching, the AIDS dataset from GED, and the L500 dataset from the Large Random Datasets. Furthermore, we compare the results with some baseline methods. GEDGNN is a method specifically designed for the AIDS dataset, and NGM is unable to handle large instances like L500 due to its modeling constraints. "Ours-$k$" refers to experiments using our model with a sample size of $k$. The following table shows the experimental results for these three datasets:
> >
> > | Method | GM-I Obj$\uparrow$ | L500 Obj$\uparrow$ | AIDS Gap$\downarrow$ | AIDS Acc(%)$\uparrow$ |
> > |-|-|-|-|-|
> > | GEDGNN | - | - | 1.515 | 42.6 |
> > | NGM | 9219.79 | - | 2.859 | 13.27 |
> > | $\Delta$-Search | 9670.62 | 29783.6 | 2.021 | 25.89 |
> > | Ours-1 | 9685.68 | 29036.4 | 1.214 | 46.76 |
> > | Ours-2 | 9693.83 | 29945.1 | 0.897 | 58.14 |
> > | Ours-4 | 9700.50 | 30781.7 | 0.489 | 73.39 |
> > | Ours-8 | 9704.76 | 31261.0 | 0.275 | 80.91 |
> > | Ours-16 | 9706.79 | 31826.5 | 0.137 | 87.84 |
> > | Ours-32 | 9707.54 | 32349.2 | 0.087 | 92.17 |
> > | Ours-64 | 9708.14 | 32848.6 | 0.065 | 93.84 |
> > | Ours-128 | 9708.35 | 33167.1 | 0.053 | 94.99 |
> >
> > From the experimental results, we observe that as the sample size increases, the model's performance improves. However, when the sample size reaches around 64 to 128, the improvement becomes less significant, indicating diminishing returns.
> >
> > Moreover, compared to the AIDS dataset, the performance improvements from increasing the sample size are less pronounced on the GM-I and L500 datasets. We believe that this is because the instances in the GED dataset are much smaller than those in the Graph Matching and Large Random datasets. As a result, **increasing the sample size allows for better exploration of the solution space, making it easier to find the optimal solution**. For larger datasets, although increasing the sample size explores a larger portion of the solution space, the search space is so vast that the performance gains are not as noticeable.
> >
> > It is also worth noting that **even with a sample size of 1, our method still shows significant performance advantages over the baselines** across all three datasets. Only on the L500 dataset does the objective with sample size 1 slightly lag behind $\Delta$-Search, but as the sample size increases, our model surpasses it.
> >
> > In conclusion, the increase in the number of samples does have an impact on performance, which depends on the characteristics of the dataset. But even with a sample size of 1, our model consistently outperforms other baselines. As stated in Section 4.1, "Sampling Tests" paragraph, "**the high quality of the solutions is primarily due to the effectiveness of BiQAP’s design, rather than an increased sampling size.**" These additional experiments will be included in the revised version.

---

> > > ### Author Response · Authors · 2024-11-18
> > > **Answer (3/3, 3 in total)**
> > >
> > > >***Q6: "The regularization strength $\epsilon$ of the inner problem is most likely an important hyperparameter (affecting the inner solution and its differentiation), it should be discussed and experimentally tested. What happens when it is set to very large or very small values?"***
> > >
> > > A6. Thank you for your suggestion! **In fact, the inner optimization of our QAP solver is based on Gromov-Sinkhorn, and its $\epsilon$ parameter serves a role similar to the temperature parameter $\tau$ in softmax (see A2-3 and A4 for Reviewer UsqU).** The setting of the $\epsilon$ hyperparameter is empirical, which is why we did not explore its impact in depth and instead set $\epsilon$ to 0.05 for all experiments. However, to demonstrate its effect on the model, we conduct related experiments as described below.
> > >
> > > | $\epsilon$ | GM-I Obj$\uparrow$ | L500 Obj$\uparrow$ | AIDS Gap$\downarrow$ | AIDS ACC$\uparrow$ |
> > > |---|:---:|:---:|:---:|---|
> > > | 0.05 | 9708.35 | 33167.1 | 0.053 | 94.99 |
> > > | 0.1 | 9708.42 | 32863.3 | 0.061 | 93.96 |
> > > | 0.2 | 9712.11 | 31127.8 | 0.105 | 90.11 |
> > > | 0.5 | 9696.43 | 27086.0 | 1.184 | 46.24 |
> > >
> > > Note that we also conduct experiments with $\epsilon$ values of 0.01 and 0.02. However, both of these configurations experience exponential explosion during training, which is caused by the exponential operation in line 6 of Algorithm 1 in our paper. However, due to normalization, $\epsilon$ values of 0.05 and above do not result in this issue. As shown in the results above, **when $\epsilon$ is too small, the model suffers from exponential explosion during training, while a larger $\epsilon$ leads to poor training convergence and reduced performance.** Therefore, we chose $\epsilon = 0.05$, a value small enough to achieve a good performance but large enough to ensure stable training.
> > >
> > >
> > > >***Q7: "No ablation for different architectures of the FormulaNet is included. It would be important to see how the SSM compares to GNN or Transformer-based architectures."***
> > >
> > > A7. Thank you for your suggestion! As mentioned in in the first paragraph of Section 3.2, we explored the potential of other architectures and their pros and cons. However, we clarified that in lines 231–233 of our revised version, "In this paper, the use of Mamba is an initial choice, leaving room for future exploration and refinement in the design of FormulaNet." Given this, **we opted for a simple FormulaNet architecture based on SSM to embed the QAP formulas, rather than overdesigning it.**
> > >
> > > We actually experimented with a Transformer-based FormulaNet earlier, but since our main emphasis was not on the design of FormulaNet itself, extensive tuning or design was not performed. **Our primary focus was on the QAP optimization framework and the differentiable QAP solver combining Gromov-Sinkhorn,** rather than on FormulaNet’s specific architecture. As a result, an ablation study comparing these different architectures was not conducted.
> > >
> > > However, we will include additional relevant experiments. As mentioned in lines 214–215 of our revised version, "GNNs are better suited for structured inputs like graphs, whereas our input matrices lack clear structural patterns." Therefore, we do not pursue a GNN-based FormulaNet but instead experiment with a Vision Transformer-based FormulaNet. Below are the experimental results on the Graph Matching dataset GM-I, comparing the performance of our designed ViT-based and SSM-based FormulaNet:
> > >
> > > | Architecture | GM-I Obj$\uparrow$ |
> > > |---|:---:|
> > > | ViT-based | 9341.59 |
> > > | SSM-based | 9708.35 |
> > >
> > > The experimental results show that the SSM-based FormulaNet outperforms the ViT-based FormulaNet. This is the reason we chose to use our designed SSM-based FormulaNet.

---

> ### Author Response · Authors · 2024-11-24
> **Friendly Reminder for Feedback Before Discussion Deadline**
>
> **Dear Reviewer,**
>
> Thank you for your valuable response. As the discussion deadline approaches, we kindly request your feedback of our response to the concerns you previously raised. We are also open to any further questions or suggestions that you may have.
>
> We sincerely appreciate your time and attention. Your guidance is truly invaluable to our research.
>
> Best regards,
>
> The Authors

---

> ### Comment · Reviewer_L8oe · 2024-11-25
> **Response to Rebuttal**
>
> Thank you for providing a thorough answer to my review. The additional experimental results resolve my corresponding concerns and will make a good addition to the manuscript.
>
> Regarding Q4 I still have a followup question. As I understand, the point the authors want to make is that visual graph matching focuses on learning the feature extraction, whereas QAP in combinatorial optimization focuses on solving individual difficult instances.
>
> I am not convinced that Figure 1 serves this purpose well, but if you really want to use it to make this point, I think you should at least explicitly state somewhere that the mismatch in Fig. 1 is a consequence of the particular choice of weighted adjacency matrix (is it euclidean distance in the particular example?), whereas in visual graph matching the weights would be given as the output of a learnable model that gets the visual features as input.

---

> > ### Author Response · Authors · 2024-11-25
> >
> > Thank you sincerely for your suggestions—they are incredibly helpful! Regarding the distinction between visual graph matching (VGM) and our QAP optimization, we agree that "visual graph matching focuses on learning the feature extraction", but we disagree with the statement “focuses on solving individual difficult instances”. The difference between our work and VGM lies in the task objectives, rather than focusing on "solving individual difficult instances."
> >
> > Here, we provide further clarification:
> >
> > - The purpose of learning in VGM and our approach is fundamentally different. VGMs are essentially visual tasks. Given supervised data for visual matching, **their goal is to use supervised learning to achieve visual matching between nodes**. In contrast, our approach is a combinatorial optimization task, where the goal is **to learn how to optimize the objective function under given constraints**. The key difference is that visually consistent matches do not necessarily correspond to the optimal objective, which is precisely the purpose of Figure 1.
> >
> > - VGM models primarily rely on manually labeled node pairs (e.g., Figure 1(a)). Their training data and evaluation metrics during testing are based on these pairs. The original node features in VGM is extracted from pre-trained VGG [1] (non-learnble), and edge features (adjacency matrix) are derived from the Euclidean distance between nodes. These VGM models mainly use GNN-based models to learn modified node features and then adopt the Sinkhorn algorithm to obtain a doubly stochastic matrix as the matching prediction between graphs.
> >
> > - Following [2], we also treat visual graph matching as a QAP problem, where the QAP parameters are derived from the extracted node and edge features using their methods. Under this setup, Figure 1(a) represents the dataset's ground truth (i.e. Supervision obtained through manual annotation), which provides visually consistent matches but does not achieve an optimal QAP objective. Figure 1(b) represents the optimal solution computed by Gurobi, which is visually inconsistent yet achieves a better objective. Therefore, **in the context of a combinatorial optimization (CO) task, Figure 1(b) is not a "mismatch"—it is correct from the perspective of the CO task.** (The issue of Figure 1 is not unique; similar in-consistency problems exist in many VGM datasets.)
> >
> > In conclusion, the purpose of Figure 1 is to highlight the difference between our paper (which frames the problem as a CO task) and previous VGM tasks. The key distinction is that VGM experiments are not fundamentally CO tasks. **This is why we do not follow VGM for experiments but instead follow [2] and [3] to generate QAP graph matching datasets for our experiments.**
> >
> > ---
> >
> > **Thus, the following statement reflects some misunderstandings:**
> >
> > > ***"The mismatch in Fig. 1 is a consequence of the particular choice of weighted adjacency matrix (is it euclidean distance in the particular example?)"***
> >
> > Whether there is a mismatch depends on the task objective. In the VGM task, Figure 1 (a) is correct, and Figure 1 (b) is incorrect. Conversely, in the combinatorial optimization task, Figure 1 (a) represents a suboptimal solution, while Figure 1 (b) is the optimal solution.
> >
> > Additionally, for VGM tasks, they usually use Euclidean distance to construct the adjacency matrix based on node features. Then in Figure 1 the node features and adjacency matrix were used to construct the QAP instance. From input images to feature extraction and then to QAP instance construction, this represents a complete pipeline provided by previous VGM approaches such as [4].
> >
> > > ***"In visual graph matching the weights would be given as the output of a learnable model"***
> >
> > In VGM, the adjacency matrix is derived from node features extracted by a pre-trained VGG model. Both the node features and the adjacency matrix are fixed and not learnable.
> >
> > ---
> >
> > Lastly, thank you again for your valuable comments! **We will incorporate this clarification into the revised version of the paper.**
> >
> > [1] Very Deep Convolutional Networks for Large-Scale Image Recognition. ICLR, 2015.
> >
> > [2] Learning Combinatorial Embedding Networks for Deep Graph Matching. ICCV, 2019.
> >
> >
> > [3] Towards Quantum Machine Learning for Constrained Combinatorial Optimization: a Quantum QAP Solver. ICML, 2023.
> >
> > [4] Neural Graph Matching Network: Learning Lawler's Quadratic Assignment Problem with Extension to Hypergraph and Multiple-graph Matching. TPAMI, 2021.

---

> > > ### Comment · Reviewer_L8oe · 2024-11-25
> > >
> > > Thank you for the fast response. I don't think there was a major misunderstanding here, but apologies for my imprecise language. What I meant by learnable vision model was the combination of the fixed VGG backbone + the learnable GNN for node feature refinement on top. This does not change my point though, as the final node features that are used to construct the QAP instance are the result of a learnable (the GNN part) model.
> > >
> > > Now, Fig 1b is the result of solving a QAP instance which is built based on node features extracted from a previous VGM model. Therefore, the reason why Fig 1b is visually inconsistent is simply because the features are not great (in the visual consistency sense), and different node features would lead to different, potentially more visually consistent QAP solutions.
> > >
> > > I guess this is also closely related to the point that the authors probably want to make here: When solving the QAP instances built from the features of previous methods one actually gets visually inconsistent results. This is because the previous methods don't actually solve the exact QAP instance, but instead run an approximate solver that in some sense "corrects" this and still returns visually consistent matchings. However, of course, when one is interested in solving actual QAP instances the previous methods are therefore of little use, and this is why a comparison to the presented method is obsolete.
> > >
> > > Thank you for adding the note that the features used to build the instance in Fig 1b were taken from a previous VGM model. I think this will make the takeaway of the figure clearer, although I still think the presentation would have room for improvement here.

---

> > > > ### Author Response · Authors · 2024-11-26
> > > >
> > > > Thank you for your valuable suggestions! Your latest understanding is indeed correct, and we appreciate the time and effort you have put into reviewing our work. We will do our best to revise the paper to make the explanation of this part (concerning Q4 and A4) clearer.
> > > >
> > > > Finally, we appreciate your positive recognition of our experiments and explanations. We are glad to know that our rebuttal has addressed most of your concerns. We kindly request your further consideration and hope for a positive outcome.

---

### Official Review · Reviewer_UsqU · 2024-11-02

**Soundness:** 3
**Presentation:** 3
**Contribution:** 2
**Rating:** 5
**Confidence:** 3

**Summary:**

This paper proposes an unsupervised learning framework, BiQAP, based on a bilevel optimization model, to solve the Koopmans-Beckmann Quadratic Assignment Problem (KBQAP). In this framework, the outer level objective corresponds to the original QAP objective, while the inner level is a differentiable Gromov-Sinkhorn QAP solver applied to new QAP instances generated by a neural network, FormulaNet. FormulaNet is trained by minimizing the original QAP objective. Extensive experiments across five tasks demonstrate its effectiveness and efficiency compared to both non-learning and learning-based methods.

**Strengths:**

1. The bilevel optimization formulation used to design an unsupervised learning framework for solving KBQAP is novel, contributing to a clearer understanding of the proposed framework's methodology.

2. The comprehensive experimental results highlight the notable effectiveness and efficiency of BiQAP in comparison to existing methods.

**Weaknesses:**

1. The core concept behind the proposed BiQAP framework appears counterintuitive. If understood correctly, the primary idea is to train a neural network that takes the original QAP instance as input and outputs a modified QAP instance. This network is trained so that, when the Gromov-Sinkhorn algorithm is applied to the generated QAP instance, the resulting solution yields a lower objective value for the original QAP. However, it is unclear why solving a modified QAP instance should yield better results than directly solving the original QAP. No discussion, theoretical analysis, or explanation is provided to justify why this approach would be effective for solving the KBQAP.

2. Although the proposed BiQAP framework is based on a bilevel optimization model, the paper lacks a review of relevant literature on bilevel optimization, as well as a discussion on why the bilevel optimization model in Eq. 3 is preferable to directly solving the original QAP.

3. This work presents a practical method but lacks theoretical analysis or discussion. For example, it does not clarify the relationship between the bilevel optimization model in Eq. 3 and the original QAP, nor does it discuss whether the solution to Eq. 3 can approximate or recover a solution to the original QAP. Additionally, the paper does not establish any properties or guarantees regarding the quality of outputs generated by the proposed BiQAP framework.

**Questions:**

1. In the numerical experiments, are the numbers of outer and inner iterations different for training and testing? If so, how crucial is this for the performance of the BiQAP?

2. Numerous existing studies focus on algorithms for solving bilevel optimization models. How does the proposed BiQAP framework relate to these studies, and could insights from this literature potentially enhance the BiQAP framework?

---

> ### Author Response · Authors · 2024-11-18
> **Answer (1/2, 2 in total)**
>
> >***Q1: "In the numerical experiments, are the numbers of outer and inner iterations different for training and testing? If so, how crucial is this for the performance of the BiQAP?"***
>
> A1. Yes, it is different. As illustrated in Section 4.1, "Training Setup" paragraph, "The number of outer and inner iterations is set to 10 and 15 during training, and 20 and 25 during testing." This configuration is based on experimental findings, which show several advantages of using fewer iterations during training and slightly more during testing:
>
> - During training, fewer iterations mitigate numerical issues such as gradient vanishing and explosion, which may occur in practical training scenarios with large iteration counts. Moreover, smaller iteration counts reduce the forward computation time of our differentiable QAP solver, enhancing training efficiency. But excessively small iterations may compromise the accuracy of the QAP solver. Based on these considerations, 10 and 15 are chosen for the outer and inner iterations, respectively.
>
> - During testing, increasing the iteration count compared to training often yields better results. In Appendix B and Table 7 of our paper, we conduct an ablation study comparing different settings for outer/inner iterations: 10/15, 20/25, and 30/35. The results indicate that the configuration of 20/25 achieves superior performance compared to the other two settings. Nevertheless, the performance differences among these results remain minimal.
>
> Overall, within a certain range, variations in outer and inner iteration counts have limited impact on the overall results.
>
> >***Q2: "Numerous existing studies focus on algorithms for solving bilevel optimization models. How does the proposed BiQAP framework relate to these studies, and could insights from this literature potentially enhance the BiQAP framework?"***
>
> >***Q3: "The paper lacks a review of relevant literature on bilevel optimization, as well as a discussion on why the bilevel optimization model in Eq. 3 (Eq. 4 in revised version) is preferable to directly solving the original QAP."***
>
> A2-A3. Thank you for pointing out the shortcomings in our paper. **In the updated version, we have added a discussion on the application of bi-level optimization in deep learning.** Here, we briefly introduce the main idea we followed, which is based on bi-level optimization to design the loss function. This forms the basis of our preparation for answering Q4.
>
> We mainly follow [1] that understanding or designing the loss via bi-level optimization:
> $$
> \min_\theta KL({Y}\mid P^\theta),
> s.t.
> {P}^\theta = \arg \min_{P1=1}  \langle  {C}^\theta,{P} \rangle- \epsilon H({P})
> $$
> where ${C}^\theta$ represents the cosine distance for the batch features with parameters $\theta$, and ${Y}$ is the known supervision for learning. As proven in [1], $H({P}) = -\langle{P}, \log {P} - {1}\rangle$ is the entropic regularization with coefficient $\epsilon$. **The inner optimization is exactly equivalent to the softmax activation, while the outer optimization corresponds to cross-entropy.** Thus we can find the above bi-level optimization equals to InfoNCE loss:
>
> $$
> \min_\theta \mathcal{L} = \sum_{i,j}Y_{ij}\log (\frac{e^{-C^\theta_{ij}/\epsilon}}{\sum_{k}e^{-C^\theta_{ik}/\epsilon}})
> $$
>
> Thus, bi-level optimization is fundamentally a method for designing activation layers or loss functions. In [1,2], modifications to the inner optimization improve the loss. ****Our work follows this learning framework, but we modify the inner optimization to use an entropic QAP, solving it with the Gromov-Sinkhorn algorithm to obtain the predicted probability matching matrix.**** The outer optimization is adjusted to use the original QAP objective as the loss, resulting in an unsupervised learning framework.
>
>
> [1] Understanding and generalizing contrastive learning from the inverse optimal transport perspective. ICML2023.
>
> [2] OT-CLIP: Understanding and generalizing clip via optimal transport. ICML2024.

---

> > ### Author Response · Authors · 2024-11-18
> > **Answer (2/2, 2 in total)**
> >
> > >***Q4: "It does not clarify the relationship between the bilevel optimization model in Eq. 3 (Eq. 4 in revised version) and the original QAP, nor does it discuss whether the solution to Eq. 3 (Eq. 4 in revised version) can approximate or recover a solution to the original QAP."***
> >
> > A4. As discussed in A3, our work does not claim that the QAP is equivalent to a bi-level optimization problem. **Instead, our bi-level optimization is essentially a loss design aimed at learning the QAP.** Specifically, in the inner optimization, we first input the original QAP parameters $\mathbf{F}_1, \mathbf{F}_2,$ and $\mathbf{K}_p$ into the neural network, to obtain a new entropic regularized QAP with parameters $\mathbf{F}_1^{\theta}, \mathbf{F}_2^{\theta},$ and $\mathbf{K}_p^{\theta}$:
> >
> > $$
> > \min_{\mathbf{X}} -\text{tr}(\mathbf{X}^\top\mathbf{F}_1^{\theta}\mathbf{X}\mathbf{F}_2^{\theta})-\text{tr}({\mathbf{K}_p^{\theta}}^\top\mathbf{X})-\epsilon H(\mathbf{X}),
> > $$
> >
> > $$
> > \text{ s.t. }\mathbf{X1}=\mathbf{1},\mathbf{X}^{\top}\mathbf{1}\leq\mathbf{1}
> > $$
> > This optimization is the inner optimization in Eq. 3 (Eq. 4 in revised version), where the Gromov-Sinkhorn algorithm (see Algorithm 1) is applied as a differentiable solver to solve the new entropic regularized QAP and obtain the solution $\mathbf{X}^\theta$. Note that the differentiable solver acts as an activation layer similar to softmax or Sinkhorn, allowing gradient backpropagation. Finally, given the calculated solution $\mathbf{X}^\theta$, we minimize the negative objective of the original QAP in the outer optimization, which serves as the loss function:
> > $$
> > \min_\theta  -\text{tr}\big(({\mathbf{X}^{\theta}})^\top\mathbf{F}_1\mathbf{X}^{\theta}\mathbf{F}_2\big)-\text{tr}(\mathbf{K}_p^\top\mathbf{X}^{\theta})
> > $$
> > Certainly, the real learning process involves the random sampling of the inner optimization, which we will not discuss further. For more specific details, please refer to Section 3.2.
> >
> > >***Q5: "It is unclear why solving a modified QAP instance should yield better results than directly solving the original QAP."***
> >
> > A5. We provide explanations from two perspectives, as detailed below:
> >
> > - **Relevant data experiments are conducted**, as shown in Appendix B Ablation Study and Table 7. In Table 7, "BiQAP w/o FN" denotes our method without FormulaNet, where our QAP solver directly takes the original instance as input. The experimental results indicate that directly solving the original QAP instance using our QAP solver yields significantly lower performance across multiple datasets compared to solving the modified QAP instance.
> >
> > - A common issue with current matrix-iteration-based algorithms is their tendency to **get stuck in local optima**, resulting in suboptimal performance. To address this, we combine the matrix-iteration algorithm of our QAP solver with neural networks by introducing learnable parameters. This approach aims to **transform instances that are prone to local optima into instances less likely to encounter such issues**. Additionally, we use multi-sampling methods for initialization, generating diverse solutions to further reduce the likelihood of getting trapped in local optima.

---

> ### Author Response · Authors · 2024-11-24
> **Friendly Reminder for Feedback Before Discussion Deadline**
>
> **Dear Reviewer,**
>
> Thank you for your valuable response. As the discussion deadline approaches, we kindly request your feedback of our response to the concerns you previously raised. We are also open to any further questions or suggestions that you may have.
>
> We sincerely appreciate your time and attention. Your guidance is truly invaluable to our research.
>
> Best regards,
>
> The Authors

---

> > ### Comment · Reviewer_UsqU · 2024-11-24
> >
> > Thank you to the authors for their rebuttal. While the numerical experiments demonstrate the efficiency of the proposed BiQAP framework, the relationship between the QAP and the bilevel optimization model remains unclear. Furthermore, it is not adequately explained why the bilevel optimization model enhances the performance of the proposed method. The paper lacks sufficient theoretical discussion on these points. Based on these considerations, I will maintain my current score.

---

> > > ### Author Response · Authors · 2024-11-25
> > > **Brief Clarification**
> > >
> > > > ***The relationship between the QAP and the bilevel optimization model remains unclear.***
> > >
> > > Thank you for your response and insights. While we have clearly explained the motivation and process of using bi-level optimization in A2-A4, covering aspects such as related work and theoretical derivation, it seems there may still be some misunderstanding. We would like to further emphasize the following points:
> > >
> > > - **Bi-level optimization is used as a method following [1] to design activation layer and loss function.** There is no strict equivalence (and indeed, it would not be feasible to establish a strict equivalence between an optimization problem involving neural networks and QAP).
> > > - In [1], bi-level optimization was used to derive the well-known InfoNCE loss. Similarly, we have followed this framework to design bi-level optimization tailored to QAP, which led to the derivation of activation and loss function integrated with Gromov Sinkhorn.
> > > - The inner optimization is effectively transformed into a differentiable solver, implemented as an activation layer, while the **outer optimization aligns with the objective of QAP**, corresponding to the loss function.
> > >
> > > We truly appreciate your time and effort in reviewing our work! We sincerely hope you will carefully review our response in detail.
> > >
> > > [1] Understanding and generalizing contrastive learning from the inverse optimal transport perspective. ICML2024.
> > >
> > > ---
> > >
> > > > ***It is not adequately explained why the bilevel optimization model enhances the performance of the proposed method.***
> > >
> > > Thank you for your question! We addressed this in A5, and together with the analysis of bi-level optimization in A2-A4, it should clarify why the bi-level optimization model enhances performance. However, it seems there may still be some misunderstanding. Let us briefly explain again:
> > >
> > > - Directly optimizing the QAP objective essentially reduces to a heuristic search problem, relying on gradient descent or other methods **without involving any neural network parameters**. As a result, the model's representational and problem-solving capabilities are inherently limited.
> > > - As explained in A2-A4, the bi-level optimization approach we use introduces neural network parameters, transforming the process of solving QAP into a loss design problem. **This enables the use of neural networks' strong fitting capabilities to tackle the QAP problem.**
> > > - In A5, we analyzed the superiority of bi-level optimization **from an experimental perspective** and further explained how the use of neural networks enhances bi-level optimization **by addressing the local optima challenges in combinatorial optimization algorithms.**
> > >
> > > We hope this explanation helps clarify your concerns!

---

> ### Author Response · Authors · 2024-11-27
> **Clarification & Kindly Request for Reconsideration**
>
> Dear Reviewer UsqU,
>
> Thank you for your time and effort! Regarding your concern about the clarity of our theoretical explanation, we would like to reiterate the following points:
>
> - In the above response and A2-A4, **we have provided a detailed explanation of "the relationship between the QAP and bilevel optimization" and introduced related work to help readers better understand the context of our approach.** This part has been updated in our revised version.
> - We would like to emphasize that **we have provided a formal proof of Eq. 6, i.e., the inner optimization, in Appendix D** (Appendix C of the original version).
> - Similarly, in the above response and A2-A5, **we have explained from multiple angles "why the bilevel optimization model enhances the performance of the proposed method."**
>
> We believe that our theoretical explanation is clear, and we have sufficiently demonstrated the effectiveness of our bilevel framework.
>
> ***We kindly ask you to reconsider our response and the revised version of the paper. If you still feel that our theoretical explanation is lacking or if there are any misunderstandings, we would appreciate it if you could raise specific questions, and we would be happy to discuss them with you.***
>
> Once again, thank you for your time and valuable insights!

---

### Author Response · Authors · 2024-11-21
**Paper Update and Reminder**

Dear Area Chair and Reviewers,

We have updated our revised PDF based on the reviewers' suggestions, with specific changes highlighted in blue. The updates we made are as follows:

- **(Reviewer UsqU, L8oe and WMpS)** To better explain the design of our bi-level framework and implicit optimization, we added related work on bi-level optimization for softmax/InfoNCE. Additionally, we elaborated on the design of our proposed bi-level framework in Section 3.1, incorporating this related work to enhance clarity for readers.
- **(Reviewer L8oe)** As mentioned in our response, we have conducted additional experiments on sampling tests.
- **(Reviewer L8oe)** Since our time metric in seconds was mentioned only once in Section 4.1 ("Evaluation" paragraph), it was easy for readers to overlook. To address this, we updated all table labels to "Time(s)" to avoid confusion.
- Minor formatting adjustments were made to improve the paper's layout.

We have addressed all comments from the reviewers and **look forward to further discussions with each reviewer**.

---

### Author Response · Authors · 2024-11-29
**General Response**

Dear PCs, SACs, ACs,

We sincerely thank all the reviewers for their valuable insights and feedback, which have significantly helped us improve our work. We have addressed each reviewer's concerns and believe we have effectively resolved most of them, receiving positive acknowledgments in return. As the author-reviewer discussion phase comes to a close, we would like to provide a general summary of the reviews and outline the efforts we have made during the discussion phase.

---

## **Reviewer Acknowledgments**

- Our bilevel optimization framework for solving KBQAP is novel and applicable (Reviewers UsqU, WMpS), offering a new paradigm for tackling non-convex and combinatorial problems (Reviewers L8oe).
- The experiments are comprehensive and demonstrate the notable effectiveness and efficiency of our approach (Reviewers UsqU, L8oe, WMpS).
- Our paper is well-structured, logical, and easy to follow (Reviewers L8oe, WMpS).

---

## **Issues Addressed in the Rebuttal**

- **Reviewer UsqU (Rating: 5)**:
  - We clearly explained the impact of the number of outer and inner iterations (A1) and why our approach achieves strong performance from multiple perspectives (A5).
  - We added relevant works on Softmax/InfoNCE to clarify the loss design and its derivation within our bilevel framework (A2-A3). We also strengthened the theoretical explanation of the relationship between our bilevel framework and the original QAP (A4). We revised the paper based on A2-A4 to improve clarity.
  - We greatly appreciate Reviewer UsqU's questions, which have helped us clarify our work. However, despite our thorough explanations, the reviewer still finds the relationship between QAP and bilevel optimization unclear (Q4), as well as the reasons why our model enhances performance (Q5), without providing specific points to highlight any shortcomings in our response. **We believe there may be a misunderstanding and would appreciate it if the reviewer could point out specific areas where our response was insufficient.** In our subsequent responses, we provided clearer and more concise explanations and are happy to engage in further discussion. However, no follow-up was provided by the reviewer.

- **Reviewer L8oe (Rating: 6)**:
    - We addressed concerns related to Gumbel initialization (A1) and provided minor clarifications (A2). Additionally, we included a theoretical analysis of the bilevel framework (A3) and expanded our sampling tests/analysis (A5). We also clarified the role and effect of the parameter $\epsilon$ (A6) and explained the selection of FormulaNet (A7).
    - Regarding A4 (the explanation of related work on visual graph matching), the reviewer provided valuable insights. After a thorough discussion, reviewer L8oe stated that while there is still room for improvement, the response was acknowledged, and the reviewer confirmed that "this will make the takeaway of the figure clearer."
    - Overall, we believe our rebuttal addressed most of the concerns raised by Reviewer L8oe, and we greatly appreciate the valuable feedback.

- **Reviewer WMpS (Rating: 6)**:

    - We provided a clearer explanation of the implicit optimization problem as requested by Reviewer WMpS. The reviewer acknowledged our clarifications, and we appreciate the reviewer's positive feedback.

---

## **Brief Summary of Our Paper**

Our paper introduces a **novel unsupervised bilevel framework** for solving the general KBQAP problem, applicable to **various combinatorial optimization tasks**. Comprehensive experiments demonstrate **strong results**, with state-of-the-art performance across multiple datasets. Our approach represents a conceptual shift, transforming multiple tasks into the KBQAP formula and optimizing it from a unified perspective. To our best knowledge, this is **the first work to optimize the KBQAP formula on these tasks**. Our success motivates us to explore similar optimization strategies for other problems, such as integer linear programming. We have also provided the code in the supplementary materials.

Best regards,

The Authors

---

### Meta-Review · Area_Chair_GDwC · 2024-12-18

**Metareview:**

The Quadratic Assignment Problem (QAP) is a computationally challenging problem with applications in areas like image matching. This paper addresses the Koopmans-Beckmann Quadratic Assignment Problem (KBQAP), and it proposed a bi-level unsupervised framework that modifies the input instance with entropic regularization, allowing for iterative solution using the Sinkhorn algorithm while preserving backpropagation.  The outer-level objective aligns with the original QAP objective, while the inner-level involves a differentiable Gromov-Sinkhorn QAP solver applied to new QAP instances generated by FormulaNet. Experimental results suggest that the new method BiQAP is effective on various benchmarks.

This paper addresses an important problem and is overall well written.  The empirical results are also promising.  However, more theoretical discussions are needed to motivate, justify, and support the bilevel optimization approach.  So far, there is little theoretical analysis, leaving it difficult to conclude the effectiveness of BiQAP.

**Additional Comments On Reviewer Discussion:**

The rebuttal has been noted by the reviewers and have been taken into account by the AC in the recommendation of acceptance/rejection.

---

### Decision · Program_Chairs · 2025-01-22

Reject